# Social corrections act as a double-edged sword by reducing the perceived accuracy of false and real news in the UK, Germany, and Italy

Florian Stoeckel [1,6]✉, Sabrina Stöckli[2,3,6], Besir Ceka[4], Chiara Ricchi[1], Ben Lyons [5] & Jason Reifler [1]

Corrective or refutational posts from ordinary users on social media have the potential to improve the online information ecosystem. While initial evidence of these social corrections is promising, a better understanding of the effects across different topics, formats, and audiences is needed. In three pre-registered experiments ($N = 1944$ UK, $N = 2467$ Italy, $N = 2210$ Germany) where respondents completed a social media post assessment task with false and true news posts on various topics (e.g., health, climate change, technology), we find that social corrections reduce perceived accuracy of and engagement with false news posts. We also find that social corrections that flag true news as false decrease perceived accuracy of and engagement with true news posts. We did not find evidence to support moderation of these effects by correction strength, anti-expert sentiments, cognitive reflection capacities, or susceptibility to social influence. While social corrections can be effective for false news, they may also undermine belief in true news.

[1] Department of Politics, University of Exeter, Exeter, UK. [2] Department of Consumer Behavior, University of Berne, Berne, Switzerland. [3] Department of Business Administration, University of Zurich, Zurich, Switzerland. [4] Department of Political Science, Davidson College, Davidson, NC, USA. [5] Department of Communication, University of Utah, Salt Lake City, UT, USA. [6]These authors contributed equally: Florian Stoeckel, Sabrina Stöckli. ✉email: F.Stoeckel@exeter.ac.uk

Some social media users engage in correcting others when they come across misinformation online, actions which are visible to an even larger group[1–4]. Most Americans not only express appreciation for these corrections, but also consider it a public responsibility[2]. Such *social corrections* (also known as observational corrections, i.e., corrective cues placed by other social media users) have been shown to be effective at preventing the spread of health-related online misinformation (e.g., ref. [5]). However, important questions about social corrections remain. Do these corrections work beyond the health context in the US? Does the effect of social corrections depend on their form and strength? Does falsely "correcting" *true* news items have similar, but given their veracity, less desirable effects? What are the underlying psychological mechanisms of social corrections, and for whom might they be more or less effective?

In this pre-registered, cross-country experimental research, we examine the effect of social corrections on the response to social media posts with *false* and *true* news in the UK, Italy, and Germany. Between July 2022 and February 2023, a total of 6621 respondents were asked to evaluate the accuracy of headlines presented in social media posts (170,220 observations) on various topics (e.g., health, climate change, technology). Each participant rated a set of nine social media posts after answering a set of pre-treatment questions. Respondents either saw social media posts without user comments (control condition) or one of three treatment conditions that included user comments, some of which denote the original post as inaccurate.

The attempts of social media platforms to directly intervene against false content may be ineffective[6], and users are often skeptical of social media platforms' top-down interventions. Nearly four out of five (78%) of US adults say that they prefer if social media platforms use people over algorithms to decide what is true or false, and two-thirds (67%) want this fact-checking to come from people of various backgrounds (i.e., different racial, ethnic, and political groups)[7]. Corrections from friends appear especially effective[8], but social media commentary may be influential regardless of any connection to the poster[9].

Increasing evidence suggests that misinformation spreads less if other social media users correct it. Seeing others add corrective cues to a post can (1) reduce how accurate one perceives this post to be, (2) reduce the probability of interacting with this post, (3) alter one's attitudes towards the post, and can (4) decrease one's intention to do what the post recommends[5,10–14]. For example, social media users are less likely to believe a post saying that the Zika outbreak in Brazil was caused by the release of genetically modified mosquitoes if others flag the post as false[5].

How broadly do these findings generalize across topics and space? Researchers testing the effect of social corrections have mostly focused on health topics with US samples[5,13,14] (for two exceptions, see e.g., refs. [10,15]). It is particularly important to examine whether such messages are effective for more contested topics (e.g., political news) that might trigger directional motivated reasoning and hence rejection, as some work suggests social corrections may be more limited in these cases[16]. Second, existing research has operationalized social corrections in different ways—from subtle, standard social media reactions (e.g., like, angry emoji) to substantiated corrective comments with links to bolstering webpages that have been reposted many times[13]. Hence, we do not know to what extent the form and strength of social correction measures determines their effect.

Third, Bode and Vraga[17] warn that social *mis*corrections of true news might amplify the spread of misinformation, at least in some cases. Specifically, they find that when social media users flag factually accurate information—e.g., tick bites can trigger an allergy to red meat—as incorrect, people are less likely to believe this information. This finding is in line with the argument that

there are no inherent differences between true and false claims themselves, and their accuracy depends on their coherence with the real world rather than built in linguistic markers – which requires audiences to bring pre-existing knowledge to bear (but see e.g. ref. [18]). Along these lines, we might expect social miscorrections to just as readily distort public understanding of facts. Clearly, we need to further probe this potential for corrections to have negative effects when they are implemented on *true* news regardless of intent.

Lastly, does the effectiveness of social corrections vary within the population? While several prominent studies have suggested that beliefs and attitudes polarize in response to (corrective) information (e.g., ref. [19]), more recent work suggests that interventions have more "parallel" effects across subgroups (e.g., refs. [20–22]). We continue this line of research by examining whether the effectiveness of social corrections varies across different segments of the population. Answering this question can tell us more about how these corrections might work, and in the process help us gain more general theoretical insight. To date, the mechanism behind their effect remains murky. The influence of social commentary may outweigh that of the professional news stories it accompanies[9], perhaps due to the primacy of social information online more generally[23–25]. Still, it remains unclear how these messages are processed. If processed centrally, argument strength and quality of evidence should matter[26]. Likewise, if these corrections are given closer scrutiny, a stronger predisposition against expert claims might lead some readers to reject them[27], while reliance on heuristics might lead to greater acceptance[28]. If social corrections are effective primarily due to the power of social norms, we might expect those most susceptible to social influence to accept them more readily[11]. We consider here the potential for heterogenous treatment effects across these psychological characteristics and orientations.

Using online experiments in the UK, Italy, and Germany, we address open questions about how and whether social corrections work to correct misinformation in various settings, formats, or across a variety of topics. We also address whether social corrections have the potential to undermine acceptance of accurate information. If this is the case, it raises profound questions about the utility of social corrections writ large. At the same time, considering how social corrections affect both accurate and inaccurate claims may yield insight into the potential mechanism of correction effects, which remains an open question.

The central goal of our research is to investigate the consistency and extent to which variations of social corrections can help stop the spread of false information on social media, and whether there is an equivalent effect when true news posts are erroneously corrected[5,9,17,29]. Our pre-registered hypotheses are as follows:

| | |
|---|---|
| $H_{Correct\ false}$ | Social corrections decrease the perceived accuracy of false news posts. |
| $H_{Correct\ false\ amplification}$ | The effects of social corrections are stronger when they include greater amplification (more "likes," multiple comments, or supporting link). |

We also hypothesize that the more amplified social corrections are, the stronger their effect[30,31]. To this end, we test different "amplification operationalizations." In the UK, we test whether social corrections are more effective when a corrective comment is liked by more (vs. only a few) individuals, whereas in Italy and Germany, we test whether social corrections have a stronger effect when more corrective comments are shown. In all three countries, we test whether social corrections are more effective if

corrective comments include a link to a fact-checking website. Note that testing different amplification operationalizations provides information about the role of the form and strength of social corrections.

We also hypothesize about corrections applied to accurate news and information, which we label *social miscorrection*. Our preregistered expectations for miscorrections are the same as for corrections – they will reduce the perceived accuracy of a claim, and effects will increase with amplification.

| | |
|---|---|
| $H_{Miscorrect\ true}$ | User comments that denote a true news post as factually incorrect (social miscorrections) decrease its perceived accuracy. |
| $H_{Miscorrect\ true\ amplification}$ | The effects of social miscorrections are stronger when they include greater amplification (more "likes," multiple comments, or supporting link). |

We also explore whether individual differences that affect people's accuracy judgements can shed light on how social correction measures work. Specifically, we identify three individual differences that might alter how people judge the accuracy of and engage with news posts on social media: anti-expert sentiments, cognitive reflection capacities, and susceptibility to social influence. More specifically, we explore the following research questions (RQs). These RQs were posed as hypotheses in the pre-registration filed for initial data collection in the UK, since we initially expected that individual level characteristics would moderate correction effects. Based on inconclusive results in the UK, these hypotheses were subsequently posed as RQs for the preregistrations filed for fieldwork in Italy and Germany.

First, anti-expert sentiments determine the extent to which people trust experts and expert knowledge, and as a consequence also contribute to their susceptibility to misinformation[27,32,33]. People who distrust experts might be less likely to take other social media users' views into account (e.g., when they link to a fact-checking site). Hence, the effect of social corrections could depend on anti-expert sentiments.

| | |
|---|---|
| $RQ_{Anti-expert}$ | Do anti-expert sentiments moderate the effect of social corrections? |

Cognitive reflection capacities determine the extent to which people engage in deliberate and effortful judgment processes when encountering new information as well as their susceptibility to misinformation[34]. People with poor (vs. strong) cognitive reflection capacities rely more on heuristics when making decision and have been shown to be particularly susceptible to nudges[28]. Thus, individuals with relatively poor cognitive reflection capacities might be more likely to be influenced by social corrections than individuals with strong cognitive reflection.

| | |
|---|---|
| $RQ_{Cognitive\ reflection}$ | Do cognitive reflection capacities moderate the effect of corrections? |

Susceptibility to social influence determines the extent to which people comply with what others do or expect of them[11,35]. Thus, individuals who are more responsive to informative and normative cues from others might also be more affected by social correction measures on social media than those who are not.

| | |
|---|---|
| $RQ_{Social\ influence}$ | Does susceptibility to social influence moderate the effect of corrections? |

## Methods
To test our hypotheses and RQs, we use three online experiments with the following structure: After answering a set of pretreatment questions, respondents assessed a random set of social media posts with *true* and *false* news on diverse topics such as health, climate change, technology, and migration. As can be seen in Fig. 1, we randomly exposed respondents to one of four conditions of each social media post: a *control* condition, *low amplification* condition, *high amplification* condition, or a *correction with link* condition. After seeing each post, respondents assessed (1) the accuracy of a post, (2) the probability of "liking" it, and (3) the probability of sharing it. After the social media post assessment task, respondents were debriefed, and provided with accurate information about the topics they saw posts about. While our hypotheses focus on perceived accuracy as outcome, we also preregistered the same hypotheses and analyses for the probability to "like", and the probability to share each post as outcomes. In this context, Epstein et al.[36]. raise an important issue. They explore potential pitfalls when asking about multiple outcomes (perceived accuracy, sharing intention) in social media studies, as we do in our design. Our primary interest is how experimentally manipulated stimuli affect these outcomes; whether and how experimental effects vary for one outcome (e.g., accuracy) based on whether or not another outcome is also asked (e.g., sharing intention) is not addressed by Epstein and colleagues[36] and remains an open question. More broadly, while Epstein et al.[36]. show differences in how people respond based on which outcomes are asked, it also remains an open issue which specific question or set of questions is the best experimental equivalent of what occurs "in the wild." We return to this important point later on.

All experiments have been pre-registered at OSF (UK: https://osf.io/fpm2e/?view_only=1f2999c931c84404bddd618ce33208bd (12 July 2022), Italy: https://osf.io/upzm8/?view_only=55393ea5c1634c2ea87c793f0cfc07d3 (12 August 2022), Germany: https://osf.io/rfq6h/?view_only=26e88087c3c442ff810b6ce452736e75 (21 January 2023). We also provide our material, data, and code on the OSF project repository; data for the UK: https://osf.io/4hjcf, for Italy: https://osf.io/yvdj4, and for Germany: https://osf.io/jhwfg. We obtained ethical approval for our research in all three countries from the Ethics Committee of the College of Social Science and International Studies at the University of Exeter in June 2022. This research complies with General Data Protection Regulation requirements. The data were collected and made available on OSF without identifying information. We obtained consent from all participants. All respondents were offered a local incentive by Dynata. Statistical tests reported below are two-sided. Our linear regression models have normally distributed residuals.

**British fieldwork**. We conducted an online experiment with a UK-sample via Dynata ($N = 1944$, 50.9% f, 48.7% m, 0.4% non-binary, third gender, and other; July, 2022). Supplementary Table 11 provides demographic details.

Our sample size was informed by a simulation-based power analysis. We aimed to be powered enough to identify a small interaction of social corrections with our proposed individual difference moderators (e.g., susceptibility to social influence, anti-expert sentiments). We used the linear mixed-effects model that we specified for the interaction of social corrections with

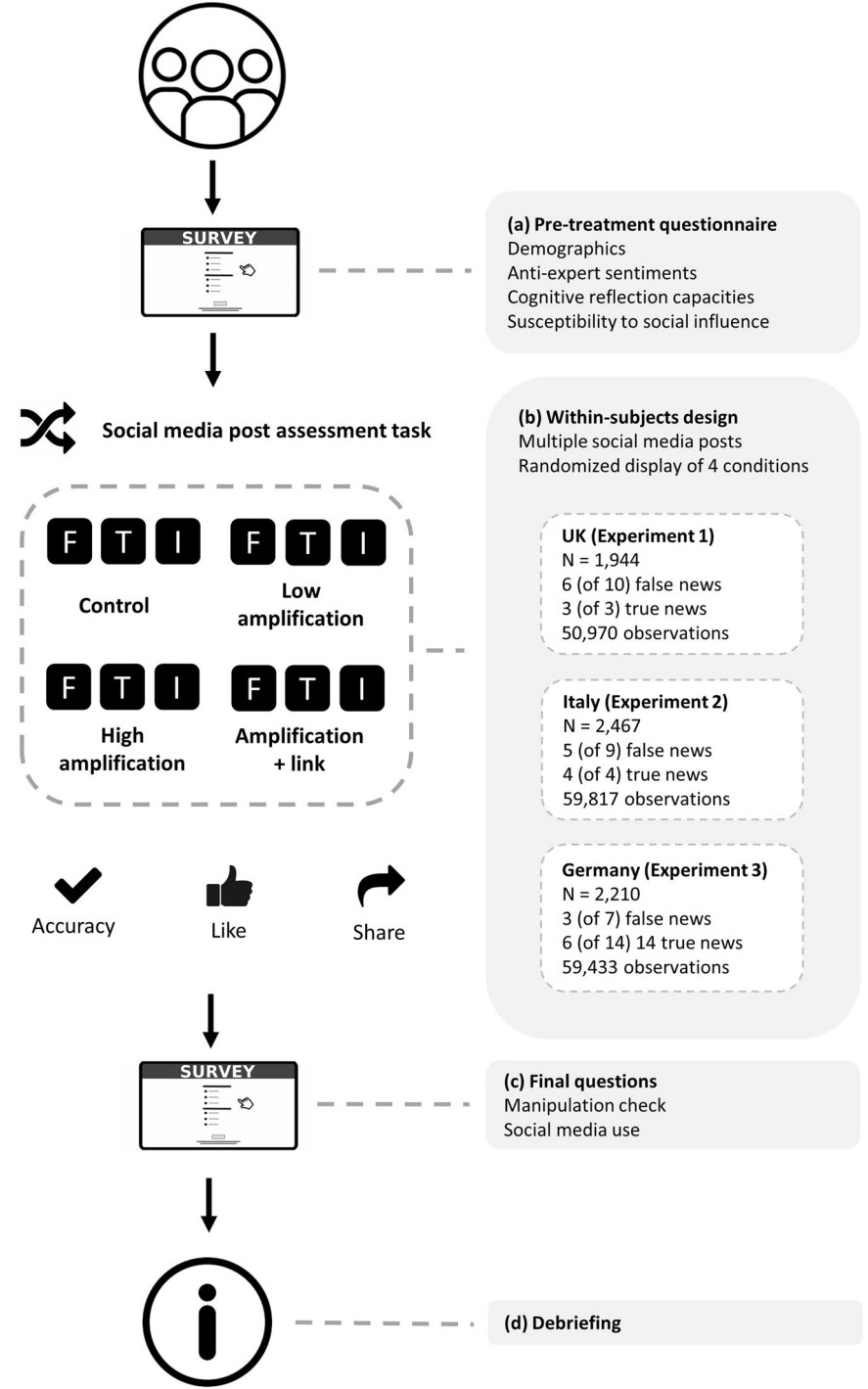

**(a) Pre-treatment questionnaire**
Demographics
Anti-expert sentiments
Cognitive reflection capacities
Susceptibility to social influence

**Social media post assessment task**

**(b) Within-subjects design**
Multiple social media posts
Randomized display of 4 conditions

Control
Low amplification
High amplification
Amplification + link

Accuracy   Like   Share

**UK (Experiment 1)**
N = 1,944
6 (of 10) false news
3 (of 3) true news
50,970 observations

**Italy (Experiment 2)**
N = 2,467
5 (of 9) false news
4 (of 4) true news
59,817 observations

**Germany (Experiment 3)**
N = 2,210
3 (of 7) false news
6 (of 14) 14 true news
59,433 observations

**(c) Final questions**
Manipulation check
Social media use

**(d) Debriefing**

susceptibility to social influence for our power analysis, which we computed based on pilot study data: perceived accuracy ~ social correction*susceptibility to social influence (perceived accuracy is modeled as repeated measures and the model contains a random intercept for the social media post). Aiming for 80% power and assuming an effect size of -0.07 for the slope between the *control* and the *high amplification + link* condition, showed that we need a sample size of $N = 1800$ (see pre-registration for details).

Prior to the experiment, respondents filled out a set of pretreatment batteries in order for us to be able to capture demographics and individual differences such as susceptibility to social influence and anti-expert sentiments (see Fig. 1). Then, respondents were given instructions for the social media post assessment task. The experiment is a within-subjects design. Every respondent was exposed to nine social media posts (repeated measures) and asked to imagine that these posts came up in their feed. Every post was randomly displayed in either the *control*, *low amplification*, *high amplification*, or *correction/miscorrection with link* condition. Six of the nine posts were false news posts, whereas the other three posts were true news posts. All participants were presented with the following sequence: (1) one true news post, (2) three false news posts (random subset of 10 posts), (3) one true news post, (4) three false news posts (random subset of 10 posts), and (5) one true news post. After seeing every post, respondents were asked to assess the (1) accuracy of, (2) probability of "liking", and (3) the

**Fig. 1 Overview of the experimental procedure and design.** All experiments had four parts: **a** participants completed a battery of pretreatment questions; **b** participants completed a task assessing randomly assigned social media content from four conditions (false news: control, a correction with low amplification, a correction with high amplification, or a correction with link; true news: control, miscorrection with low amplification, miscorrection with high amplification, or a miscorrection with link). For each post, respondents reported its perceived accuracy, the probability of "liking" it, and the probability of sharing the post. Next, **c** participants were asked a set of final questions and were **d** debriefed. The debriefing debunked false information shown in posts. While the control condition was the same in all countries for both false and true news posts (no comments), the operationalizations for the treatment conditions varied slightly across countries. UK: low amplification: one correction/miscorrection comment with few "likes"; high amplification: one correction/miscorrection comment with many "likes"; corrections/miscorrections with link: includes weblink that refers to a site that debunks false news (in case of false news) or a link to a site that bolsters miscorrection (in case of true news). Italy: low amplification: one corrective and one supporting comment (for false and true news); high amplification: multiple corrective comments (for false news) or multiple miscorrective comments (for true news) among supporting comments; correction with link: resembles the low amplification condition, except that the correction (false news) and miscorrection (true news) include links to websites that support the respective statements. Germany: operationalizations for false news posts resemble those implemented in Italy, albeit the correction with link condition is similar to the high amplification condition, with one corrective statement including a link to a fact-checking website. For true news posts, we used the following operationalization: a miscorrection with high amplification (multiple miscorrective and one supporting comment); a miscorrection with high amplification, albeit with a source cue (logo of media outlet) that indicates the source of the information in the original post, and a miscorrection with link that bolsters the inaccurate statement (the condition resembles the high amplification condition without source cue in other respects). F Facebook, T Twitter (now X), I Instagram.

probability of sharing the post. Thus, there is randomization but also a fixed sequence (in all countries) which ensures that all respondents see false as well as true news. Next, respondents filled out some sociodemographic information, were debriefed, and were provided with debunking information on the posts.

*Social media post assessment task.* We created a set of social media posts that includes ten false ones and three true ones. The post content was taken from real-world social media posts. For all posts, there are either websites that officially debunk content (for false news) or "bolster" content with information (for true news). For each of the posts, we either created a standardized Facebook or Twitter post using an open-source social media simulator (https://zeoob.com/). Every post was created in four conditions: (1) in the *control* condition, posts did not contain any comments; (2) in the *low amplification* condition, posts included a corrective comment in the case of false news (e.g., 'Well, this is a blatant lie') as well as a supporting comment (e.g., 'There is something seriously wrong when the 'cure' [vaccines] kills more children than the 'problem.') or a miscorrection in the case of true news. These comments had up to fifteen "likes"; (3) in the *high amplification* condition, posts were identical to the low amplification condition, but the corrective or miscorrective comment had more likes (between 100 and 200), (4) in the *correction/miscorrection with link* condition, posts were identical to the high amplification condition, but additionally included a fact-checking link (e.g., from https://fullfact.org/) below the comment (see Figs. 2 and 3; for all posts see https://osf.io/4hjcf, folder "posts" in "methods"). The link led to a website that bolstered a miscorrection in the case of true news.

To ensure that respondents perceived a difference between the low and high amplification condition, we added a manipulation check question after the experimental task. That is, we showed a comment (e.g. "You know this isn't true. Or even remotely close to the truth") with ten "likes" (*low amplification*) and the same comment with 184 "likes" (*high amplification*) and asked respondents to indicate how strongly they perceive the support from other users (1 = *very weak*; 5 = *very strong*). We find support for our operationalization: respondents reported that the comment from the *high amplification* (vs. *low amplification*) condition received statistically significantly more support from other users (Ms (SDs)= 3.02 (1.15) vs. 2.77 (1.13); $t = 6.17$, $p < 0.001$). We increased the external validity of the content used in the fieldwork by showing user comments that were actually written and posted by users on social media (albeit they might be from other posts than the ones shown). As a result, some

corrections are more directly indicating that an original post is substantively inaccurate than others. For instance, the comment that there is "something deeply deeply fishy about this" might be perceived as a correction or in fact as agreement with the original content. Most comments are direct and explicit corrections of the original post, e.g.: "This post is disingenuous if not a straight forward lie". (This is a corrective comment in False News Stimulus 8 from the UK; see osf for all materials).

We did not use the original names of the people who posted the stories that we use, but generated names with an online random name simulator (https://britishsurnames.co.uk/random). We did not use the original profile pictures but used pictures from the Flickr-Faces-HQ Dataset (FFHQ)[37]. Moreover, we created a picture with the original claim and used them as main post pictures for our stories. We kept platform-specific settings constant across the different posts and conditions (e.g., no location tags, online user status, no 'own' reaction to post).

After being exposed to every post, we asked three questions: The first question captured perceived accuracy: *'To the best of your knowledge, how accurate is the claim in the above headline?'* (1 = *not at all accurate*; 4 = *very accurate*). The second question captured the probability of liking the post: *'How likely are you to "like" the article/post in the headline that you just read?'* (1 = *not at all likely*; 5 = *very likely*). The third question captured the probability of sharing the post: *'How likely are you to share the article/post in the headline that you just read?'* (1 = *not at all likely*; 5 = *very likely*).

*Anti-expert sentiments.* To capture anti-expert sentiments, we used a 3-item battery (e.g., *'I am more confident in my opinion than other people's facts.'*) with a 5-point scale (1 = *strongly disagree*, 5 = *strongly agree*; Han et al.,[27] Uscinski et al.[38]). The reliability was satisfactory ($\alpha = 0.76$). For our analyses, we use the mean score. Higher scores indicate stronger anti-expert sentiments.

*Cognitive reflection capacities.* To capture respondents' cognitive reflection, we used four validated Cognitive Reflection Test (CRT) items (e.g., *'If it takes 10 machines 10 minutes to make 10 objects, how long will it take 70 machines to make 70 objects?'*). For every item, we provided four standard options, including the intuitive-incorrect and the correct response[39,40]. For our analyses, we computed the sum score of correct responses for every respondent. Higher CRT sum scores indicate more cognitive reflection capacities.

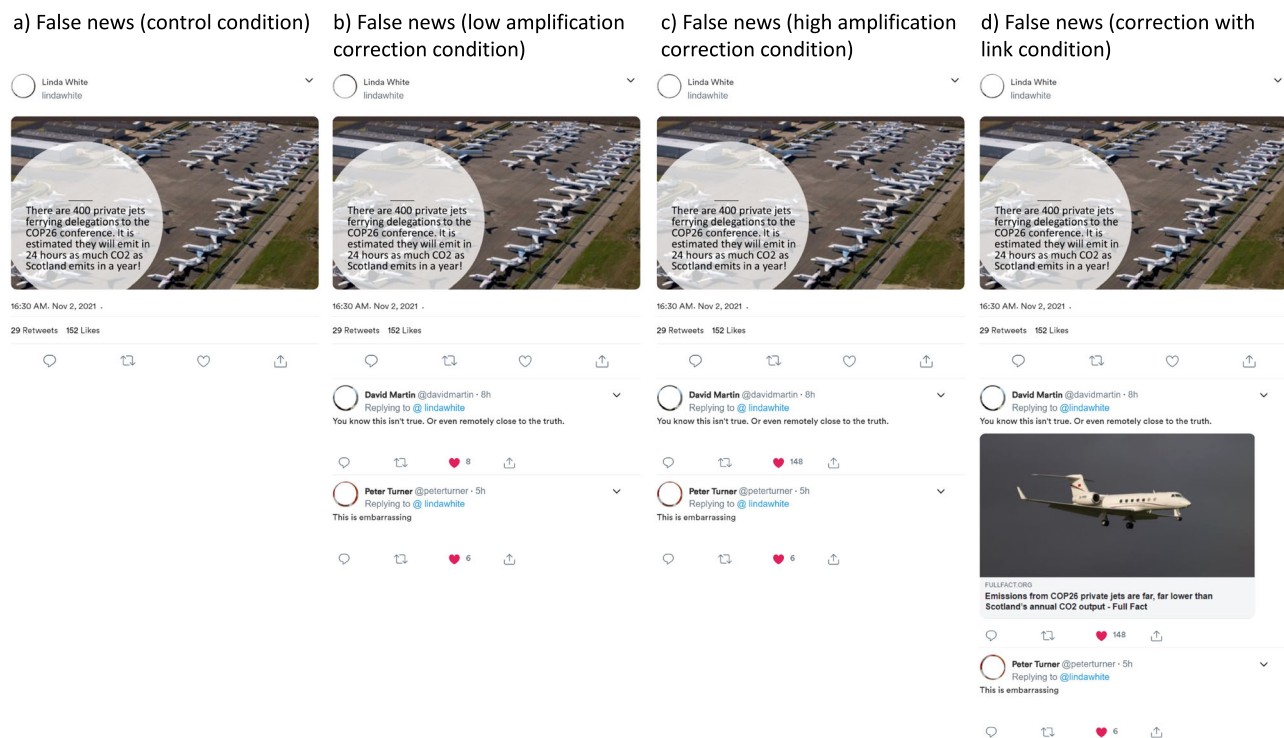

**Fig. 2 Examples of false news posts showing the four conditions in the UK.** Figure 2a: The false news control condition includes no comments. Figure 2b: The false news low amplification condition includes a corrective comment with few likes (less than 15). Figure 2c: The false news high amplification correction condition includes a corrective comment with many likes (more than 100) in the UK. In Italy and Germany, high amplification refers to several corrective comments being shown (rather than many likes). Figure 2d: The false news correction with link condition includes a corrective comment with a link to a fact-check which debunks the false news of the original post. All names are randomly generated and are not real names. Profile photos were removed at the request of the publisher. Information shown in the post was debunked in a debriefing and is available here: https://fullfact.org/environment/cop26-private-jets-scotland-carbon-emissions-year/ Information that debunks all false news used in this study can be found in Supplementary Table 36.

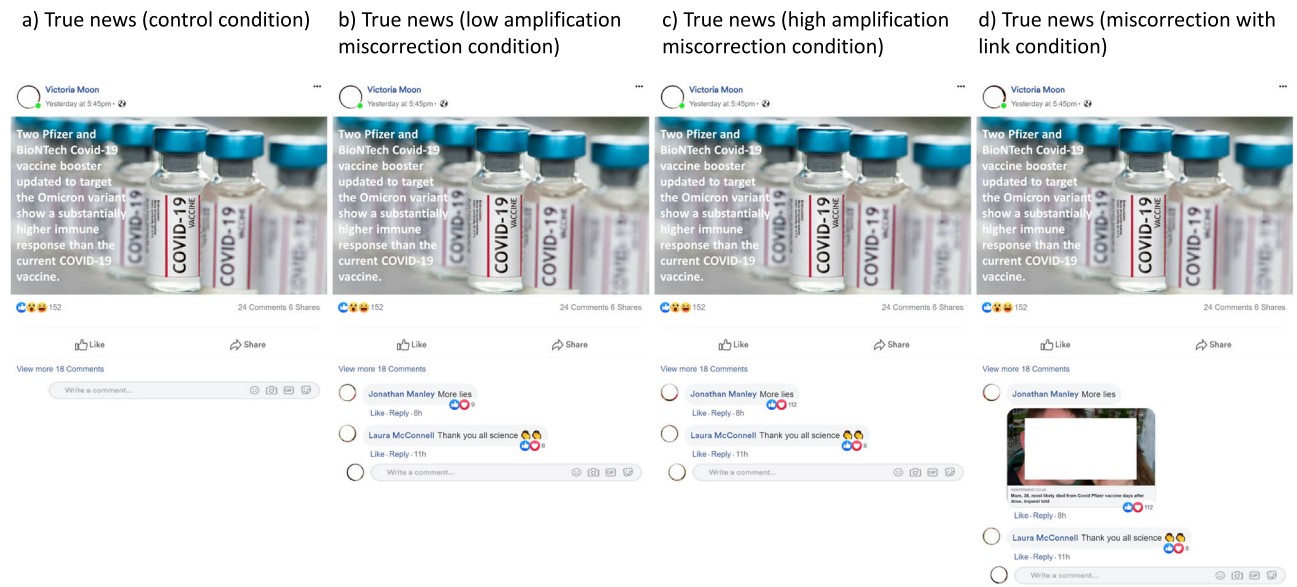

**Fig. 3 Examples of true news posts showing the four conditions in the UK.** Figure 3a: The true news control condition includes no comments. Figure 3b: The true news low amplification condition includes a miscorrection comment with few likes (less than 15). Figure 3c: The true news high amplification condition includes a miscorrection comment with many likes (more than 100) in the UK. In Italy and Germany, high amplification refers to several miscorrection comments being shown instead of a high number of likes (see files on osf for the complete treatment material). Figure 3d: The true news miscorrection with link condition includes a comment with a link to a website that seemingly bolsters a miscorrection. All names are randomly generated and are not real names. Profile photos were removed at the request of the publisher.

*Susceptibility to social influence.* To capture susceptibility to social influence, we used a 7-item battery (e.g., *'I use posts on online social networks to help me make better decisions.'*) with a 5-point scale (1 = *strongly disagree*, 5 = *strongly agree*; Stöckli et al.[41],). The reliability was satisfactory (α = 0.95). For our analyses, we use the mean score. Higher scores indicate stronger susceptibility to social influence.

Here we provide an overview of the main linear mixed-effects regression models (in lme4 notation) reported (for details see our preregistration, https://osf.io/fpm2e/?view_only=1f2999c931c844 04bddd618ce33208bd). For the main effect of social corrections on the three response variables of interest—perceived accuracy, the probability of "liking", and the probability of sharing false news—we entered the social correction treatment (4-level factor) as main predictor into the model. We also added demographics as covariates. The random structure was specified by entering a random intercept for respondents and social media posts.

$$\text{Response} \sim \text{social correction treatment} + \text{gender} + \text{age} + \text{education}$$
$$+ (1|\text{respondent id}) + (1|\text{social media post})$$
$$(1)$$

To test interactions of social correction with individual differences—namely anti-expert sentiments, CRT, and SSI—we computed additional models by entering the respective individual differences measure and their interaction with the social correction treatment into the model:

$$\text{Response} \sim \text{social correction treatment} * \text{individual difference measure}$$
$$+ \text{gender} + \text{age} + \text{education} + (1|\text{respondent id}) + (1|\text{social media post})$$
$$(2)$$

**Italian fieldwork**. We conducted an online experiment with an Italian sample via Dynata ($N = 2,467$, 49.1% f, 50.5% m, 0.3% non-binary, third gender, and other; July, 2022). Supplementary Table 22 provides demographic details. Like in the UK Experiment, our sample size was informed by a simulation-based power analysis, which suggested that we need a sample size of $N = 1800$ (see our preregistration for details).

The procedure and design were largely the same as in the UK (see Fig. 1). The social media assessment task exposed every respondent to nine social media posts (repeated measures). Similar to the UK fieldwork, real and false news stimuli were presented in a set order: 1 real followed by 2 false (randomly selected from 9 possible posts), followed by 1 real and 1 false (randomly selected from 9 possible posts), followed by another real story and another 2 false (again randomly selected from 9 possible posts), with one final real story.

*Social media post assessment task.* We created a set of nine false and four true news social media posts. Again, we used real-world social media content, and created standardized Facebook or Instagram posts using the same open-source social media simulator as in the UK experiment.

The main deviation from the experiment that we conducted in the UK was the operationalization of the social correction conditions. The operationalization for the *control* condition remained the same, i.e., showed no comments. For the social media posts with false news, the three treatment conditions were operationalized as follows: *Low amplification* = posts included one corrective and one supporting comment, *high amplification* = posts included multiple corrective and one supporting comment, *correction with link* = posts were identical to the low amplification condition, but additionally included a fact-checking link below the corrective comment at the top (all post are available on osf at https://osf.io/yvdj4/). For the social media posts with true news, the three treatment conditions were operationalized as follows: *Low*

*amplification* = posts included one miscorrective and one comment that supported the original post, *high amplification* = posts included multiple miscorrections and two supporting comments, *miscorrection with link* = posts were identical to the low amplification condition, but additionally included a link to a website that bolsters the miscorrection.

Just like for the experiment in the UK, we generated names with a random name simulator (https://www.fantasyname generators.com/italian-names.php), used profile pictures from the FFHQ database[37], created a story picture with the original claim, and kept platform-specific settings constant across the different posts and conditions (e.g., no location tags). The perceived accuracy, the probability of "liking", and sharing posts were measured with the same questions as in the UK experiment.

*Individual differences measures.* To capture respondents' cognitive reflection capacities, susceptibility to social influence, and anti-expert sentiments, we used the same items as in the UK (Anti-expert sentiments: α = 0.68, Susceptibility to social influence: α = 0.94).

*Analysis plan.* We computed the same analyses as in the UK. For details see our preregistration; https://osf.io/upzm8/?view_only=55393ea5c1634c2ea87c793f0cfc07d3).

**German fieldwork**. We conducted an online experiment with a German sample via Dynata ($N = 2,210$, 50.3% f, 49.4% m, 0.3% non-binary, third gender, and other; January/February 2023). Supplementary Table 33 provides demographic details. Like in the previous experiments, our sample size was informed by a simulation-based power analysis, which suggested that we need a sample size of $N = 1,800$ (for details see our preregistration).

**Procedure and design**. The procedure and design were largely the same as in the previous experiments (see Fig. 1). The main deviation from the UK was that the social media assessment task exposed every respondent to three false and six real social media posts (repeated measures). As in the British and Italian fieldwork, the order of post veracity was set in advance in Germany. Respondents received 3 real news posts (out of 14 possible), then 3 false (out of 7 possible), and then 3 more real (out of 14 possible).

*Social media post assessment task.* We created a set of seven false and 14 true news social media posts. Again, we used real-world social media content, and created standardized Facebook, Instagram, and Twitter posts using the same open-source social media simulator as in the previous experiments.

The operationalization of the social correction conditions for false news posts was the same as in our previous experiment in Italy. The only difference when it comes to false news is that the "correction with link" condition is similar to the high amplification condition, but additionally includes a link to a fact-checking website. The main deviation from the Italian fieldwork was the operationalization of the social correction conditions for true news posts. For the social media posts with true news, the three treatment conditions were operationalized as follows: *high amplification (without source cue)* = multiple miscorrective and one supporting comment; *high amplification (with source cue)* = same as the low amplification condition, but with a source cue to a media outlet displayed in the original post; *miscorrection with link* = same as high amplification condition, but with one miscorrective comment showing a link to a bolstering website (all post are available on osf at https://osf.io/jhwfg/).

As before, we generated names with a random name simulator (https://fossbytes.com/tools/random-name-generator), used profile pictures from the FFHQ database[37], created a story picture with the original claim, and kept platform-specific settings constant across the different posts and conditions (e.g., no location tags). The perceived accuracy, the probability of "liking" and sharing posts were measured with the same questions as in the previous experiments.

*Individual differences measures.* To capture respondents' cognitive reflection capacities, susceptibility to social influence, and anti-expert sentiments, we used the same items as in the fieldwork in Italy (Anti-expert sentiments: $\alpha = 0.76$, Susceptibility to social influence: $\alpha = 0.95$). We computed the same analyses as for the British experiment. For details see our preregistration; https://osf.io/rfq6h/?view_only=26e88087c3c442ff810b6ce452736e75).

**Reporting summary**. Further information on research design is available in the Nature Portfolio Reporting Summary linked to this article.

## Results

**Social corrections are effective at tackling *false* news**. To test $H_{Correct\ false}$, we conducted a linear mixed-effects regression analysis with perceived accuracy for false news posts as the response (repeated measures), and social correction as predictor (see methods for model details). Our results broadly support our hypothesis (see Fig. 4). All correction conditions reduced perceived accuracy in the UK (*low amplification*: $B = -0.10$, $SE = 0.02$, $t(9833) = -5.13$, $p < 0.001$, 95% CI $[-0.13, -0.06]$; *high amplification*: $B = -0.13$, $SE = 0.02$, $t(9833) = -6.69$, $p < 0.001$, 95% CI $[-0.16, -0.09]$; *correction with fact-checking link*: $B = -0.11$, $SE = 0.02$, $t(9833) = -6.04$, $p < 0.001$, 95% CI $[-0.15, -0.08]$) and in Germany (*low amplification*: $B = -0.10$, $SE =$

$0.03$, $t(6587) = -3.85$, $p < 0.001$, 95% CI $[-0.15, -0.05]$; *high amplification*: $B = -0.14$, $SE = 0.03$, $t(6587) = -5.42$, $p < 0.001$, 95% CI $[-0.19, -0.09]$; *correction with fact-checking link*: $B = -0.16$, $SE = 0.03$, $t(6587) = -5.98$, $p < 0.001$, 95% CI $[-0.21, -0.10]$). In Italy, the *high amplification* and *correction with fact-checking link* condition reduced the perceived accuracy (*high amplification*: $B = -0.12$, $SE = 0.02$, $t(10186) = -6.31$, $p < 0.001$, 95% CI $[-0.16, -0.08]$; *correction with fact-checking link*: $B = -0.12$, $SE = 0.02$, $t(10186) = -6.15$, $p < 0.001$, 95% CI $[-0.15, -0.08]$), but the *low amplification* correction did not have an effect ($t(10186) < |2|$, $p = 0.07$). Note that we tested for differences between conditions, but these tests do not reveal a pattern of statistically significant and coherent differences (see Supplementary Table 34 for detailed statistics for all permutations).

Besides perceived accuracy, we also measured the respondents' probability of "liking" and sharing the posts. As Fig. 4 shows, the results with these response variables are similar to the results for perceived accuracy, implying that social corrections can also reduce engagement with false news. The estimates for the random effects imply that the effect does not vary much across topics (Tables 1–3).

Note that we pre-registered all our experiments and performed robustness checks by excluding respondents that failed either pre-treatment attention check and by controlling for congeniality of post content. We provide more details on these robustness checks in the Supplementary Material; they did not lead to substantially different findings (Supplementary Methods and Supplementary Tables 7–10 for the UK, Supplementary Tables 18–21 for Italy, and Supplementary Tables 29–32 for Germany).

**Social miscorrections taint *true* news**. To test $H_{Miscorrect\ true}$, we computed the same linear mixed-effects regressions as for $H_{Correct\ false}$, but for posts with true news. Our results largely

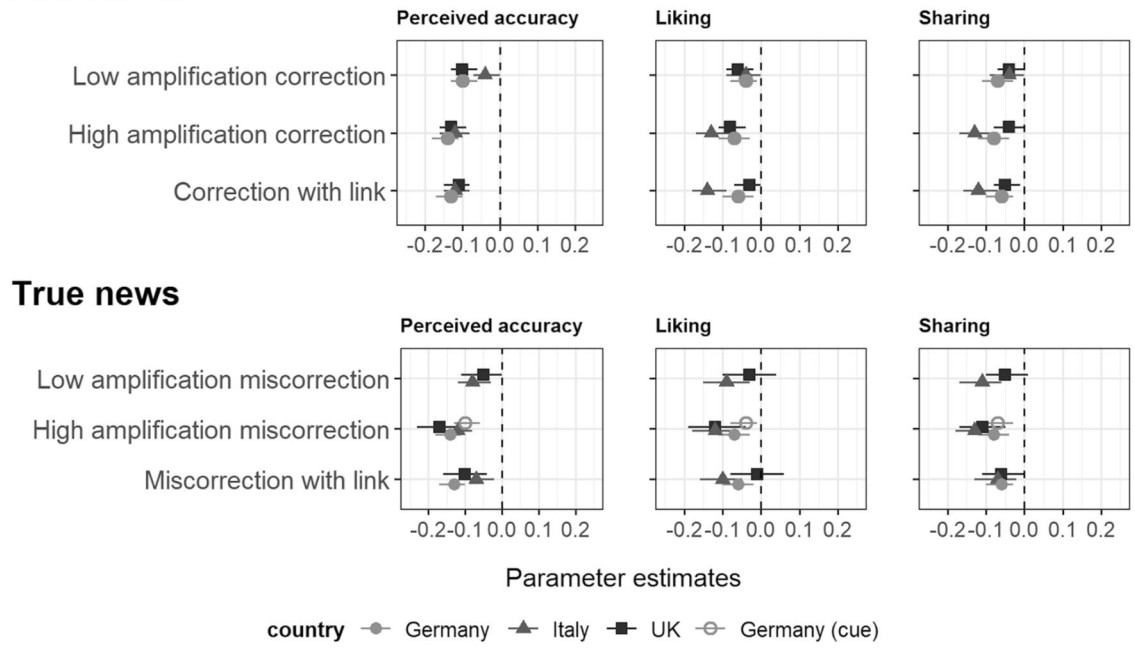

**Fig. 4 Results for false and true news for the UK, Italy and Germany.** Top panel (false news) shows fixed-effects estimates for social correction effects computed by our linear mixed-effects models used to test $H_{Correct\ false}$ in the UK, Italy, and Germany. Bottom panel (true news) shows fixed-effects estimates for miscorrection effects computed by our linear mixed-effects models used to test $H_{Miscorrect\ true}$ in the UK, Italy, and Germany. Reference group for corrections and miscorrections is the control condition (no user comments). Estimates (B) with 95% CI show how a respective correction and miscorrection condition decreases the perceived accuracy of, probability to "like", and probability to "share" false and true news posts.

**Table 1 Results of linear mixed-effects models for H$_{Correct\ false}$ in the UK.**

**Perceived accuracy**

| Fixed effects | B | t | 95% CI | p |
|---|---|---|---|---|
| (Intercept) | 2.45 | 31.38 | 2.30-2.60 | <0.001 |
| Treatment (low amplification) | −0.10 | −5.13 | −0.13-−0.06 | <0.001 |
| Treatment (high amplification) | −0.13 | −6.69 | −0.16-−0.09 | <0.001 |
| Treatment (correction with link) | −0.11 | −6.04 | −0.15-−0.08 | <0.001 |
| Gender (male) | 0.10 | 3.24 | 0.04-0.17 | 0.001 |
| Age (35-44) | 0.04 | 0.72 | −0.08-0.16 | 0.469 |
| Age (45-54) | −0.05 | −0.89 | −0.17-0.06 | 0.373 |
| Age (55-64) | −0.47 | −8.03 | −0.58- −0.35 | <0.001 |
| Age (65-75) | −0.63 | −10.52 | −0.75-−0.52 | <0.001 |
| Age (76+) | −0.70 | −12.31 | −0.81-−0.59 | <0.001 |
| Education (technical or vocational degree) | 0.00 | 0.09 | −0.07-0.08 | 0.925 |
| Education (university degree) | 0.02 | 0.51 | −0.05-0.09 | 0.611 |
| **Random effects** | **Variance** | **SD** | **mR$^2$** | 0.11 |
| Participants (intercept) | 0.40 | 0.63 | cR$^2$ | 0.59 |
| False News Posts (intercept) | 0.03 | 0.19 | | |
| Residual | 0.37 | 0.61 | | |

**Liking**

| Fixed effects | B | t | 95% CI | p |
|---|---|---|---|---|
| (Intercept) | 2.54 | 32.10 | 2.39-2.70 | <0.001 |
| Treatment (low amplification) | −0.06 | −3.03 | −0.09-−0.02 | 0.002 |
| Treatment (high amplification) | −0.08 | −4.13 | −0.11-−0.04 | <0.001 |
| Treatment (correction with link) | −0.03 | −1.76 | −0.07-0.00 | 0.078 |
| Gender (male) | 0.20 | 4.28 | 0.11-0.30 | <0.001 |
| Age (35-44) | 0.08 | 0.83 | −0.10-0.26 | 0.408 |
| Age (45-54) | −0.17 | −1.99 | −0.35-−0.00 | 0.046 |
| Age (55-64) | −0.86 | −9.95 | −1.04-−0.69 | <0.001 |
| Age (65-75) | −1.09 | −12.08 | −1.26-−0.91 | <0.001 |
| Age (76+) | −1.23 | −14.52 | −1.40-−1.07 | <0.001 |
| Education (technical or vocational degree) | 0.01 | 0.23 | −0.10-0.13 | 0.816 |
| Education (university degree) | 0.08 | 1.34 | −0.03-0.19 | 0.180 |
| **Random effects** | **Variance** | **SD** | **mR$^2$** | 0.17 |
| Participants (intercept) | 0.99 | 0.99 | cR$^2$ | 0.78 |
| False News Posts (intercept) | 0.01 | 0.08 | | |
| Residual | 0.36 | 0.60 | | |

**Sharing**

| Fixed effects | B | t | 95% CI | p |
|---|---|---|---|---|
| (Intercept) | 2.53 | 31.44 | 2.37-2.68 | <0.001 |
| Treatment (low amplification) | −0.04 | −2.13 | −0.07-−0.00 | 0.033 |
| Treatment (high amplification) | −0.04 | −2.18 | −0.08 -−0.00 | 0.029 |
| Treatment (correction with link) | −0.05 | −2.48 | −0.08-−0.01 | 0.013 |
| Gender (male) | 0.23 | 4.73 | 0.13-0.33 | <0.001 |
| Age (35-44) | 0.09 | 0.94 | −0.10-0.27 | 0.347 |
| Age (45-54) | −0.21 | −2.33 | −0.38-−0.03 | 0.020 |
| Age (55-64) | −0.93 | −10.49 | −1.10-−0.76 | <0.001 |
| Age (65-75) | −1.13 | −12.26 | −1.31-−0.95 | <0.001 |
| Age (76+) | −1.30 | −14.99 | −1.47-−1.13 | <0.001 |
| Education (technical or vocational degree) | −0.00 | −0.05 | −0.12-0.11 | 0.961 |
| Education (university degree) | 0.09 | 1.60 | −0.02-0.20 | 0.110 |
| **Random effects** | **Variance** | **SD** | **mR$^2$** | 0.18 |
| Participants (intercept) | 1.03 | 1.02 | cR$^2$ | 0.80 |
| False News Posts (intercept) | 0.01 | 0.08 | | |
| Residual | 0.33 | 0.58 | | |

The mixed-effects regressions were run on 9848 observations (perceived accuracy), 9852 observations (like), and 9851 observations (share) with 1927 respondents and 10 false news posts. mR$^2$ = marginal R$^2$ (i.e., variance of the fixed effects); cR$^2$ = conditional R$^2$ (i.e., variance of the fixed and random effects). Reference groups: Treatment = control condition; gender = female; age = 18-24; education = less than primary education.

support our hypothesis. In the UK, the *high amplification mis-correction* ($B = −0.17$, $SE = 0.03$, $t(5794) = −5.79$, $p < 0.001$, 95% CI [−0.23, −0.11]), and the *miscorrection with fact-checking link miscorrection* ($B = −0.10$, $SE = 0.03$, $t(5794) = −3.25$, $p < 0.001$, 95% CI [−0.16, −0.04]), but not the *low amplification miscorrection* ($B = −0.05$, $SE = 0.03$, $t(5794) = −1.82$, $p = 0.069$, 95% CI [−0.11, 0.00]) decreased perceived accuracy of true news posts. All miscorrection conditions reduced the perceived

accuracy in Italy (*low amplification*: $B = −0.08$, $SE = 0.02$, $t(9761) = −3.58$, $p = 0.001$, 95% CI [−0.12-−0.03]; *high amplification*: $B = −0.12$, $SE = 0.02$, $t(9761) = −5.71$, $p < 0.001$, 95% CI [−0.17-−0.08]; *miscorrection with link*: $B = −0.07$, $SE = 0.02$, $t(9761) = −3.34$, $p < 0.001$, 95% CI [−0.11-−0.02]) and in Germany (*high amplification (with source cue)*: $B = −0.10$, $SE = 0.02$, $t(13192) = −5.72$, $p < 0.001$, 95% CI [−0.13-−0.06]; *high amplification (without source cue)*: $B = −0.14$, $SE = 0.02$,

**Table 2 Results of linear mixed-effects models for H$_{Correct\ false}$ in Italy.**

**Perceived accuracy**

| Fixed effects | B | t | 95% CI | p |
|---|---|---|---|---|
| (Intercept) | 2.04 | 33.17 | 1.92–2.16 | <0.001 |
| Treatment (low amplification) | −0.03 | −1.8 | −0.07–0.00 | 0.072 |
| Treatment (high amplification) | −0.12 | −6.3 | −0.16–−0.08 | <0.001 |
| Treatment (correction with link) | −0.12 | −6.13 | −0.15–−0.08 | <0.001 |
| Gender (male) | −0.01 | −0.53 | −0.07–0.04 | 0.593 |
| Age (35-44) | 0.02 | 0.33 | −0.09–0.12 | 0.738 |
| Age (45-54) | −0.1 | −1.96 | −0.19–−0.00 | 0.05 |
| Age (55-64) | −0.06 | −1.14 | −0.16–0.04 | 0.255 |
| Age (65-75) | −0.12 | −2.32 | −0.23–−0.02 | 0.021 |
| Age (76 +) | −0.3 | −1.4 | −0.72–0.12 | 0.16 |
| Education (technical or vocational degree) | −0.07 | −1.88 | −0.15–0.00 | 0.06 |
| Education (university degree) | −0.13 | −3.6 | −0.19–−0.06 | <0.001 |
| Random effects | Var. | SD | mR$^2$ | 0.01 |
| Participants (intercept) | 0.38 | 0.62 | cR$^2$ | 0.52 |
| False News Posts (intercept) | 0.01 | 0.10 | | |
| Residual | 0.37 | 0.61 | | |

**Liking**

| Fixed effects | B | t | 95% CI | p |
|---|---|---|---|---|
| (Intercept) | 2.15 | 29.78 | 2.01–2.29 | <0.001 |
| Treatment (low amplification) | −0.04 | −1.76 | −0.09–0.00 | 0.079 |
| Treatment (high amplification) | −0.13 | −5.55 | −0.17–−0.08 | <0.001 |
| Treatment (correction with link) | −0.14 | −5.99 | −0.18–−0.09 | <0.001 |
| Gender (male) | −0.08 | −2.09 | −0.15–−0.01 | 0.036 |
| Age (35-44) | 0.01 | 0.1 | −0.13–0.15 | 0.917 |
| Age (45-54) | −0.08 | −1.22 | −0.22–0.05 | 0.224 |
| Age (55-64) | −0.08 | −1.22 | −0.22–0.05 | 0.224 |
| Age (65-75) | −0.22 | −3.09 | −0.37–−0.08 | 0.002 |
| Age (76+) | −0.36 | −1.25 | −0.94–0.21 | 0.213 |
| Education (technical or vocational degree) | −0.1 | −1.86 | −0.21–0.01 | 0.063 |
| Education (university degree) | −0.14 | −2.87 | −0.23–−0.04 | 0.004 |
| Random effects | Var. | SD | mR$^2$ | 0.01 |
| Participants (intercept) | 0.76 | 0.87 | cR$^2$ | 0.59 |
| False News Posts (intercept) | 0.01 | 0.10 | | |
| Residual | 0.54 | 0.73 | | |

**Sharing**

| Fixed effects | B | t | 95% CI | p |
|---|---|---|---|---|
| (Intercept) | 2.1 | 28.12 | 1.95–2.24 | <0.001 |
| Treatment (low amplification) | −0.04 | −1.69 | −0.08–0.01 | 0.09 |
| Treatment (high amplification) | −0.13 | −5.74 | −0.17–−0.08 | <0.001 |
| Treatment (correction with link) | −0.12 | −5.26 | −0.16–−0.07 | <0.001 |
| Gender (male) | −0.09 | −2.39 | −0.17–−0.02 | 0.017 |
| Age (35-44) | 0.01 | 0.13 | −0.14–0.15 | 0.899 |
| Age (45-54) | −0.08 | −1.21 | −0.22–0.05 | 0.226 |
| Age (55-64) | −0.08 | −1.12 | −0.22–0.06 | 0.261 |
| Age (65-75) | −0.2 | −2.75 | −0.35–−0.06 | 0.006 |
| Age (76+) | −0.32 | −1.07 | −0.91–0.27 | 0.287 |
| Education (technical or vocational degree) | −0.08 | −1.38 | −0.19–0.03 | 0.168 |
| Education (university degree) | −0.08 | −1.69 | −0.18–0.01 | 0.091 |
| Random effects | Var. | SD | mR$^2$ | 0.01 |
| Participants (intercept) | 0.81 | 0.90 | cR$^2$ | 0.62 |
| False News Posts (intercept) | 0.01 | 0.10 | | |
| Residual | 0.51 | 0.71 | | |

The mixed-effects regressions were run on 10,143 observations (perceived accuracy), 10,151 observations (like), and 10,144 observations (share) with 2445 respondents and nine false news posts. mR$^2$ = marginal R$^2$ (i.e., variance of the fixed effects); cR$^2$ = conditional R$^2$ (i.e., variance of the fixed and random effects). Reference groups: Treatment = control condition; gender = female; age = 18–24; education = less than primary education.

$t(13192) = -7.76$, $p < 0.001$, 95% CI [−0.18–−0.11]; *correction with link*: $B = -0.13$, $SE = 0.02$, $t(13192) = -7.39$, $p < 0.001$, 95% CI [−0.17–−0.10]). We also tested differences between conditions. We do not find a pattern of consistent statistically significant differences (see Supplementary Table 34 for full details for every permutation). In line with the pattern for false news, we did not find evidence that effects vary significantly across post topics.

In line with the pattern for false news, effects vary little across post topics. The reason for showing source cues in the original posts in Germany is to test if respondents use it as a heuristic for accuracy that makes miscorrections irrelevant (even if mainstream news can be inaccurate as well). Yet, the treatment effects show that miscorrections affect users despite source cues.

As Fig. 4 (bottom panel) shows, the results for the probability of "liking" and sharing posts as outcomes are similar to the results

**Table 3 Results of linear mixed-effects models for H$_{\text{Correct false}}$ in Germany.**

**Perceived accuracy**

| Fixed effects | B | t | 95% CI | p |
|---|---|---|---|---|
| (Intercept) | 2.51 | 31.61 | 2.36–2.67 | <0.001 |
| Treatment (low amplification) | −0.10 | −3.87 | −0.15–−0.05 | <0.001 |
| Treatment (high amplification) | −0.14 | −5.43 | −0.19–−0.09 | <0.001 |
| Treatment (correction with link) | −0.16 | −6.00 | −0.21–−0.10 | <0.001 |
| Gender (male) | −0.01 | −0.35 | −0.07–0.05 | 0.724 |
| Age (25–34) | −0.04 | −0.70 | −0.17–0.08 | 0.485 |
| Age (35–44) | −0.09 | −1.59 | −0.20–0.02 | 0.112 |
| Age (45–54) | −0.16 | −2.73 | −0.27–−0.04 | 0.006 |
| Age (55–64) | −0.26 | −4.58 | −0.37–−0.15 | <0.001 |
| Age (65+) | −0.29 | −4.74 | −0.41–−0.17 | <0.001 |
| Education (technical or vocational degree) | −0.14 | −3.29 | −0.23–−0.06 | 0.001 |
| Education (university degree) | −0.22 | −4.78 | −0.31–−0.13 | <0.001 |
| Random effects | Var. | SD | mR$^2$ | 0.02 |
| Participants (intercept) | 0.35 | 0.59 | cR$^2$ | 0.48 |
| False News Posts (intercept) | 0.02 | 0.14 | | |
| Residual | 0.43 | 0.66 | | |

**Liking**

| Fixed effects | B | t | 95% CI | p |
|---|---|---|---|---|
| (Intercept) | 2.62 | 31.71 | 2.46–2.78 | <0.001 |
| Treatment (low amplification) | −0.10 | −3.58 | −0.16–−0.05 | <0.001 |
| Treatment (high amplification) | −0.09 | −3.33 | −0.15–−0.04 | 0.001 |
| Treatment (correction with link) | −0.13 | −4.82 | −0.19–−0.08 | <0.001 |
| Gender (male) | 0.14 | 3.30 | 0.06–0.23 | 0.001 |
| Age (25–34) | −0.05 | −0.53 | −0.22–0.13 | 0.595 |
| Age (35–44) | −0.24 | −2.98 | −0.40–−0.08 | 0.003 |
| Age (45–54) | −0.49 | −6.09 | −0.65–−0.33 | <0.001 |
| Age (55–64) | −0.64 | −7.84 | −0.79–−0.48 | <0.001 |
| Age (65+) | −0.58 | −6.68 | −0.75–−0.41 | <0.001 |
| Education (technical or vocational degree) | −0.40 | −6.17 | −0.53–−0.28 | <0.001 |
| Education (university degree) | −0.25 | −4.04 | −0.37–−0.13 | <0.001 |
| Random effects | Var. | SD | mR$^2$ | 0.06 |
| Participants (intercept) | 0.84 | 0.92 | cR$^2$ | 0.67 |
| False News Posts (intercept) | 0.00 | 0.00 | | |
| Residual | 0.46 | 0.68 | | |

**Sharing**

| Fixed effects | B | t | 95% CI | p |
|---|---|---|---|---|
| (Intercept) | 2.68 | 31.78 | 2.52–2.85 | <0.001 |
| Treatment (low amplification) | −0.08 | −3.16 | −0.13–−0.03 | 0.002 |
| Treatment (high amplification) | −0.08 | −3.24 | −0.13–−0.03 | 0.001 |
| Treatment (correction with link) | −0.12 | −4.62 | −0.17–−0.07 | <0.001 |
| Gender (male) | 0.18 | 4.01 | 0.09–0.27 | <0.001 |
| Age (25–34) | −0.10 | −1.06 | −0.28–0.08 | 0.29 |
| Age (35–44) | −0.30 | −3.48 | −0.46–−0.13 | <0.001 |
| Age (45–54) | −0.60 | −7.17 | −0.77–−0.44 | <0.001 |
| Age (55–64) | −0.76 | −9.04 | −0.93–−0.60 | <0.001 |
| Age (65+) | −0.74 | −8.22 | −0.91–−0.56 | <0.001 |
| Education (technical or vocational degree) | −0.28 | −4.29 | −0.40–−0.15 | <0.001 |
| Education (university degree) | −0.42 | −6.21 | −0.56–−0.29 | <0.001 |
| Random effects | Var. | SD | mR$^2$ | 0.08 |
| Participants (intercept) | 0.95 | 0.97 | cR$^2$ | 0.74 |
| False News Posts (intercept) | 0.00 | 0.00 | | |
| Residual | 0.38 | 0.62 | | |

The mixed-effects regressions were run on 6,602 observations (perceived accuracy), 6604 observations (like), and 6604 observations (share) with 2202 respondents and four false news posts. mR$^2$ = marginal R$^2$ (i.e., variance of the fixed effects); cR$^2$ = conditional R$^2$ (i.e., variance of the fixed and random effects). Reference groups: Treatment = control condition; gender = female; age = 18–24; education = less than primary education.

for perceived accuracy. Thus, user comments that raise doubts about the veracity of a post can also reduce engagement with true news (Tables 4–6). We also run regression analyses that pool false and true news data to test the effect of (mis-)corrections by interacting treatment with veracity (true vs. false) (Supplementary Table 35). We do not find evidence for effects to vary in a statistically significant way depending on veracity.

**No evidence for anti-expert sentiments, cognitive reflection, and social influence as moderators**. To explore whether individual characteristics condition the effectiveness of exposure to social corrections, we tested whether anti-expert sentiments, cognitive reflection capacities, or susceptibility to social influence moderate the effect of social corrections on perceived accuracy of news posts. Overall, we do not find a systematic pattern of

**Table 4 Results of linear mixed-effects models for H$_{\text{Miscorrect true}}$ news in the UK.**

**Perceived accuracy**

| Fixed effects | B | t | 95% CI | p |
|---|---|---|---|---|
| (Intercept) | 2.55 | 52.54 | 2.45-2.64 | <0.001 |
| Treatment (low amplification) | −0.05 | −1.82 | −0.11-0.00 | 0.069 |
| Treatment (high amplification) | −0.17 | −5.79 | −0.23-−0.11 | <0.001 |
| Treatment (correction with link) | −0.10 | −3.25 | −0.16-−0.04 | 0.001 |
| Gender (male) | 0.09 | 3.14 | 0.03-0.15 | 0.002 |
| Age (25-34) | 0.09 | 1.62 | −0.02-0.20 | 0.106 |
| Age (35-44) | 0.06 | 1.18 | −0.04-0.17 | 0.239 |
| Age (45-54) | −0.09 | −1.60 | −0.19-0.02 | 0.109 |
| Age (55-64) | −0.12 | −2.25 | −0.23-−0.02 | 0.025 |
| Age (65+) | −0.09 | −1.74 | −0.19-0.01 | 0.081 |
| Education (technical or vocational degree) | 0.01 | 0.18 | −0.06-0.08 | 0.857 |
| Education (university degree) | 0.11 | 3.28 | 0.05-0.18 | 0.001 |
| Random effects | Var. | SD | mR$^2$ | 0.02 |
| Participants (intercept) | 0.21 | 0.46 | cR$^2$ | 0.30 |
| Residual | 0.53 | 0.72 | | |

**Liking**

| Fixed effects | B | t | 95% CI | p |
|---|---|---|---|---|
| (Intercept) | 2.64 | 33.79 | 2.48-2.79 | <0.001 |
| Treatment (low amplification) | −0.03 | −0.87 | −0.10-0.04 | 0.383 |
| Treatment (high amplification) | −0.12 | −3.20 | −0.19-−0.04 | 0.001 |
| Treatment (correction with link) | −0.01 | −0.35 | −0.08-0.06 | 0.723 |
| Gender (male) | 0.19 | 3.99 | 0.10-0.29 | <0.001 |
| Age (25-34) | 0.03 | 0.35 | −0.15-0.21 | 0.728 |
| Age (35-44) | −0.15 | −1.72 | −0.33-0.02 | 0.085 |
| Age (45-54) | −0.75 | −8.57 | −0.93-−0.58 | <0.001 |
| Age (55-64) | −0.89 | −9.83 | −1.07-−0.72 | <0.001 |
| Age (65+) | −0.97 | −11.32 | −1.14-−0.80 | <0.001 |
| Education (technical or vocational degree) | 0.01 | 0.10 | −0.11-0.12 | 0.921 |
| Education (university degree) | 0.06 | 1.05 | −0.05-0.17 | 0.295 |
| Random effects | Var. | SD | mR$^2$ | 0.11 |
| Participants (intercept) | 0.85 | 0.92 | cR$^2$ | 0.60 |
| Residual | 0.69 | 0.83 | | |

**Sharing**

| Fixed effects | B | t | 95% CI | p |
|---|---|---|---|---|
| (Intercept) | 2.58 | 32.68 | 2.43-2.74 | <0.001 |
| Treatment (low amplification) | −0.05 | −1.53 | −0.10-0.01 | 0.125 |
| Treatment (high amplification) | −0.11 | −3.69 | −0.17-−0.05 | <0.001 |
| Treatment (correction with link) | −0.06 | −1.85 | −0.11-0.00 | 0.065 |
| Gender (male) | 0.22 | 4.50 | 0.13-0.32 | <0.001 |
| Age (25-34) | 0.08 | 0.87 | −0.10-0.27 | 0.386 |
| Age (35-44) | −0.21 | −2.28 | −0.38-−0.03 | 0.023 |
| Age (45-54) | −0.88 | −9.81 | −1.06-−0.71 | <0.001 |
| Age (55-64) | −1.02 | −10.90 | −1.20-−0.83 | <0.001 |
| Age (65+) | −1.21 | −13.68 | −1.38-−1.03 | <0.001 |
| Education (technical or vocational degree) | 0.03 | 0.51 | −0.09-0.15 | 0.610 |
| Education (university degree) | 0.13 | 2.20 | 0.01-0.24 | 0.028 |
| Random effects | Var. | SD | mR$^2$ | 0.16 |
| Participants (intercept) | 0.98 | 0.99 | cR$^2$ | 0.73 |
| Residual | 0.46 | 0.68 | | |

The mixed-effects regressions were run on 5808 observations (perceived accuracy), 5806 observations (like), and 9805 observations (share) with 1927 respondents and three true news posts. mR$^2$ = marginal R$^2$ (i.e., variance of the fixed effects); cR$^2$ = conditional R$^2$ (i.e., variance of the fixed and random effects). Reference groups: Treatment = control condition; gender = female; age = 18-24; education = less than primary education.
As we showed the same three true news posts to all our respondents, we simplified the structure of our mixed-effects regressions by removing the random intercept for social media posts. That is, we computed the following model: response - social correction treatment + gender + age + education + (1|respondent id).

statistically significant interactions. Specifically, there is little evidence to suggest that these individual characteristics moderate the effect of user comments (corrections/miscorrections) on perceived accuracy of false news (see Supplementary Fig. 1) or true news (see Supplementary Fig. 2). Moreover, differences in anti-expert sentiments, cognitive reflection capacities, and susceptibility to social influence do not moderate the effect of social corrections on the probability of "liking" and sharing posts with either false news or true news (Supplementary Methods; UK:

Supplementary Tables 1–10, Italy: Supplementary Tables 12–21; Germany: Supplementary Tables 23–32).

## Discussion

We examine the effects of social corrections and miscorrections across three countries, employing 47 total news story stimuli covering a wide variety of topics. We find corrective cues placed by other social media users effectively reduce the perceived accuracy of and engagement with *false* news posts. Moreover, we

**Table 5 Results of linear mixed-effects models for H$_{\text{Miscorrect true}}$ in Italy.**

**Perceived accuracy**

| Fixed effects | B | t | 95% CI | p |
|---|---|---|---|---|
| (Intercept) | 2.67 | 21.66 | 2.42-2.91 | <0.001 |
| Treatment (low amplification) | −0.08 | −3.58 | −0.12-−0.03 | <0.001 |
| Treatment (high amplification) | −0.12 | −5.71 | −0.17-−0.08 | <0.001 |
| Treatment (miscorrection with link) | −0.07 | −3.34 | −0.11-−0.03 | 0.001 |
| Gender (male) | −0.03 | −1.38 | −0.08-0.01 | 0.167 |
| Age (25–34) | −0.18 | −3.95 | −0.27-−0.09 | <0.001 |
| Age (35–44) | −0.28 | −6.43 | −0.36-−0.19 | <0.001 |
| Age (45–54) | −0.27 | −6.16 | −0.35-−0.18 | <0.001 |
| Age (55–64) | −0.29 | −6.31 | −0.38-−0.20 | <0.001 |
| Age (65+) | −0.11 | −0.61 | −0.47-0.24 | 0.542 |
| Education (technical or vocational degree) | 0.11 | 3.2 | 0.04-0.18 | 0.001 |
| Education (university degree) | 0.08 | 2.69 | 0.02-0.14 | 0.007 |
| Random effects | Var. | SD | mR$^2$ | 0.02 |
| Participants (intercept) | 0.23 | 0.48 | cR$^2$ | 0.38 |
| False News Posts (intercept) | 0.05 | 0.22 | | |
| Residual | 0.48 | 0.69 | | |

**Liking**

| Fixed effects | B | t | 95% CI | p |
|---|---|---|---|---|
| (Intercept) | 2.73 | 17.76 | 2.43-3.04 | <0.001 |
| Treatment (low amplification) | −0.09 | −3.11 | −0.14-−0.03 | 0.002 |
| Treatment (high amplification) | −0.12 | −4.35 | −0.18-−0.07 | <0.001 |
| Treatment (miscorrection with link) | −0.10 | −3.76 | −0.16-−0.05 | <0.001 |
| Gender (male) | −0.06 | −1.70 | −0.14-0.01 | 0.089 |
| Age (25–34) | −0.22 | -3.09 | −0.36-−0.08 | 0.002 |
| Age (35–44) | -0.34 | −4.96 | −0.47-−0.20 | <0.001 |
| Age (45–54) | −0.35 | −5.03 | −0.48-−0.21 | <0.001 |
| Age (55–64) | −0.38 | −5.31 | -0.53-−0.24 | <0.001 |
| Age (65+) | 0.09 | 0.32 | −0.47-0.66 | 0.748 |
| Education (technical or vocational degree) | 0.10 | 2.14 | 0.01-0.19 | 0.032 |
| Education (university degree) | 0.07 | 1.36 | −0.03-0.18 | 0.175 |
| Random effects | Var. | SD | mR$^2$ | 0.01 |
| Participants (intercept) | 0.68 | 0.82 | cR$^2$ | 0.50 |
| False News Posts (intercept) | 0.08 | 0.28 | | |
| Residual | 0.77 | 0.88 | | |

**Sharing**

| Fixed effects | B | t | 95% CI | p |
|---|---|---|---|---|
| (Intercept) | 2.54 | 21.52 | 2.31-2.77 | <0.001 |
| Treatment (low amplification) | −0.11 | −4.13 | −0.16-−0.06 | <0.001 |
| Treatment (high amplification) | −0.13 | −4.88 | −0.18-−0.08 | <0.001 |
| Treatment (miscorrection with link) | −0.07 | −2.79 | −0.13-−0.02 | 0.005 |
| Gender (male) | −0.09 | −2.27 | −0.17-−0.01 | 0.023 |
| Age (25–34) | −0.17 | −2.24 | −0.31-−0.02 | 0.025 |
| Age (35–44) | −0.24 | −3.41 | −0.38-−0.10 | 0.001 |
| Age (45–54) | −0.23 | −3.28 | −0.37-−0.09 | 0.001 |
| Age (55–64) | −0.28 | −3.7 | −0.42-−0.13 | <0.001 |
| Age (65+) | 0.07 | 0.22 | −0.52-0.65 | 0.827 |
| Education (technical or vocational degree) | 0.01 | 0.25 | −0.10-0.12 | 0.799 |
| Education (university degree) | 0.12 | 2.37 | 0.02-0.21 | 0.018 |
| Random effects | Var. | SD | mR$^2$ | 0.01 |
| Participants (intercept) | 0.76 | 0.87 | cR$^2$ | 0.54 |
| False News Posts (intercept) | 0.04 | 0.20 | | |
| Residual | 0.69 | 0.83 | | |

The mixed-effects regressions were run on 9775 observations (perceived accuracy), 9773 observations (like), and 9831 observations (share) with 2445 respondents and four true news posts.
mR$^2$ = marginal R$^2$ (i.e., variance of the fixed effects); cR$^2$ = conditional R$^2$ (i.e., variance of the fixed and random effects). Reference groups: Treatment = control condition; gender = female; age = 18-24; education = less than primary education.

find no consistent statistically significant evidence that this varies by the form or strength of the corrective cues nor do we find statistically significant evidence that the effect is moderated by people's cognitive reflection capacities, levels of distrust in experts, or susceptibility to social influence.

Our research advances the growing body of literature on social correction measures[5,13–15]. First, we provide evidence on the effectiveness of social corrections outside the US and across a wide variety of topics. Second, by replicating the negative effect of "miscorrective" cues on perceived accuracy in the context of true news posts, we support Bode and Vraga's[17] warning that social correction measures in the context of true news can amplify the spread of misinformation. Third, we provide insights into the underlying mechanisms of social corrections. Overall, then, our research not only contributes to our understanding of the generalizability of social correction, but also has theoretical and policy implications.

**Table 6 Results of linear mixed-effects models for H$_{Miscorrect\ true}$ in Germany.**

**Perceived accuracy**

| Fixed effects | B | t | 95% CI | p |
|---|---|---|---|---|
| (Intercept) | 2.62 | 36.11 | 2.48–2.76 | <0.001 |
| Treatment (high amplification without cue) | −0.14 | −7.74 | −0.18–−0.11 | <0.001 |
| Treatment (high amplification with cue) | −0.10 | −5.26 | −0.13–−0.06 | <0.001 |
| Treatment (miscorrection with link) | −0.13 | −7.39 | −0.17–−0.10 | <0.001 |
| Gender (male) | 0.03 | 1.32 | −0.01–0.07 | 0.188 |
| Age (25–34) | 0.03 | 0.60 | -0.06–0.12 | 0.546 |
| Age (35–44) | 0.00 | 0.01 | −0.08–0.08 | 0.996 |
| Age (45–54) | 0.00 | −0.02 | −0.08–0.08 | 0.984 |
| Age (55–64) | 0.00 | −0.05 | −0.08–0.08 | 0.963 |
| Age (65+) | −0.03 | −0.71 | −0.12–0.05 | 0.480 |
| Education (technical or vocational degree) | 0.08 | 2.39 | 0.01–0.14 | 0.017 |
| Education (university degree) | 0.12 | 3.61 | 0.05–0.19 | <0.001 |
| Random effects | Var. | SD | mR² | 0.01 |
| Participants (intercept) | 0.18 | 0.42 | cR² | 0.32 |
| False News Posts (intercept) | 0.05 | 0.22 | | |
| Residual | 0.48 | 0.69 | | |

**Liking**

| Fixed effects | B | t | 95% CI | p |
|---|---|---|---|---|
| (Intercept) | 2.67 | 31.53 | 2.50–2.83 | <0.001 |
| Treatment (high amplification without cue) | −0.07 | −3.29 | −0.11–−0.03 | 0.001 |
| Treatment (high amplification with cue) | −0.04 | −2.23 | −0.08–−0.01 | 0.026 |
| Treatment (miscorrection with link) | −0.06 | −3.18 | −0.10–−0.02 | 0.001 |
| Gender (male) | 0.19 | 4.64 | 0.11–0.27 | <0.001 |
| Age (25–34) | −0.03 | -0.36 | −0.19–0.13 | 0.716 |
| Age (35–44) | −0.25 | −3.23 | −0.40–−0.10 | 0.001 |
| Age (45–54) | −0.47 | −6.16 | −0.62–−0.32 | <0.001 |
| Age (55–64) | −0.50 | −6.58 | −0.65–−0.35 | <0.001 |
| Age (65+) | −0.48 | −5.87 | −0.64–−0.32 | <0.001 |
| Education (technical or vocational degree) | −0.20 | −3.40 | −0.31–−0.08 | 0.001 |
| Education (university degree) | −0.31 | −5.07 | −0.43–−0.19 | <0.001 |
| Random effects | Var. | SD | mR² | 0.04 |
| Participants (intercept) | 0.79 | 0.89 | cR² | 0.60 |
| False News Posts (intercept) | 0.02 | 0.14 | | |
| Residual | 0.58 | 0.76 | | |

**Sharing**

| Fixed effects | B | t | 95% CI | p |
|---|---|---|---|---|
| (Intercept) | 2.69 | 31.67 | 2.52–2.85 | <0.001 |
| Treatment (high amplification without cue) | −0.08 | −4.41 | −0.12–−0.04 | <0.001 |
| Treatment (high amplification with cue) | −0.07 | −3.87 | −0.11–−0.03 | <0.001 |
| Treatment (miscorrection with link) | −0.06 | −3.44 | −0.10–−0.03 | 0.001 |
| Gender (male) | 0.23 | 5.35 | 0.15–0.31 | <0.001 |
| Age (25–34) | −0.10 | −1.17 | −0.27–0.07 | 0.242 |
| Age (35–44) | −0.33 | −4.17 | −0.49–−0.18 | <0.001 |
| Age (45–54) | −0.60 | −7.53 | −0.76–−0.44 | <0.001 |
| Age (55–64) | −0.67 | −8.47 | −0.83–−0.52 | <0.001 |
| Age (65+) | −0.68 | −8.02 | −0.85–−0.51 | <0.001 |
| Education (technical or vocational degree) | −0.21 | −3.44 | −0.33–−0.09 | 0.001 |
| Education (university degree) | −0.31 | −4.78 | −0.43–−0.18 | <0.001 |
| Random effects | Var. | SD | mR² | 0.06 |
| Participants (intercept) | 0.88 | 0.94 | cR² | 0.68 |
| False News Posts (intercept) | 0.02 | 0.14 | | |
| Residual | 0.47 | 0.69 | | |

The mixed-effects regressions were run on 13,207 observations (perceived accuracy), 13,208 observations (like), and 13,208 observations (share) with 2203 respondents and four false news posts. mR² = marginal R² (i.e., variance of the fixed effects); cR² = conditional R² (i.e., variance of the fixed and random effects). Reference groups: Treatment = control condition; gender = female; age = 18–24; education = less than primary education.

A particularly noteworthy result for applications of our findings is that we did not find evidence that more sophisticated corrective cues (i.e., corrective comments with links to fact-checking websites) are consistently and in a statistically significant way more effective than weak corrective cues (e.g., several words that flag a post as inaccurate). In short, the format and strength of corrective comments does not matter much. This highlights the practical feasibility of social correction in the context of *false* news posts. Social media users do not need to write long, substantiated comments to flag false content. This is important because it implies a low bar for participation in social correction[13].

The simplicity of creating effective corrections is however a double-edged sword. Social media environments also include users that flag *true* news as false, and these instances can exacerbate confusion in the public sphere. Although skepticism

towards reputable news can also be important[42], recent work has highlighted the fact that not all skepticism is healthy and can cross a line into cynicism or anti-expert sentiment[42,43]. Accordingly, given the trade-off between desirable and undesirable effects of social correction measures, it is important to reflect on the relative volume of false content shared and the accuracy of "corrective" commentary. The vast majority of news shared on social media comes from high credibility sources, rather than dubious domains[44]. Other descriptive work suggests that for at least one news story (President Trump's approval rating in 2017), up to 20% of accompanying social commentary was false[9]. It should be concerning how easily comments that raise doubts about legitimate news can affect people's accuracy judgements, especially given that a vast share of people overestimates their ability to differentiate between true and false news[45]. This part of our results highlights that the need for people to "stop and think" more often when sharing news[46] also applies to reading other users' comments.

We do not find statistically significant evidence showing that people who tend to distrust (versus trust) experts – and who might therefore reject typical fact-checkers – to be less likely to take other social media users' views into account. Non-experts as the "corrective" source might be perceived as credible by people with expert aversion—a pattern that can make social correction measures particularly useful when targeting people vulnerable to misinformation. Moreover, our results suggest that people that differ in their cognitive reflection capacities do not differ in how attentive to social corrections they are. This is contrary to the classical judgment and decision-making literatures which argue that alterations in the choice architecture ("nudges") are more likely to change the behavior of people who tend to stick to their intuitive responses rather than engage in reflection[47,48]. One explanation for this pattern is that our corrective cues are so obvious that they do not bypass anyone's conscious awareness. Hence, our corrective cues might work as prompts, i.e., conscious reminders, that work equally well for people with strong and weak cognitive reflection capacities. An alternative explanation can be found in critics of the CRT (the measure capturing cognitive reflection capacities), namely that it does not capture cognitive reflection, but rather the disposition to comply with the implicit recommendations encoded in the CRT items, an approach that does not always indicate a lack of cognitive reflection capacities[49]. Finally, we also find that people's degree of susceptibility to informative and normative cues from other social media users does not determine the extent to which they are affected by social correction. Importantly, our experimental treatment effects point in the same direction: If the social correction effects were driven by social influence, we would expect highly amplified social corrections to be more effective than less amplified ones.

Overall, our results imply that social correction measures do not necessarily trigger complex argument scrutiny, norm compliance, or social group identification mechanisms. Instead, the general efficacy of these messages may derive from their appearance as follow-up negations. Some work on debunking suggests order matters, with follow-up correction being more effective[14,50]. In the case of the social corrections tested here, the persuasive impact of recency effects seems to outweigh primacy effects in low-motivation settings[51].

**Limitations**. Our results come with limitations. The outcome variables are based on self-reports that have been recorded in an experimental setting. From an ecological validity perspective, it seems desirable to run our social media post assessment task in the field, while individuals browse their social media accounts. While ethical considerations make it impossible to experimentally run our social media assessment task on actual social media platforms, future research can analyze large-scale observational data on social corrections and miscorrections[8]. Clearly, such a correlational approach also poses substantial data-gathering and analysis challenges.

Additionally, our design asks respondents about both accuracy and sharing intent. Importantly, Epstein et al[36]. found that participants were worse at discerning truth from falsehood if they both evaluated accuracy and indicated their sharing intention. However, while Epstein et al. demonstrate that measurement properties vary based on whether one or both outcomes are asked, there is not a strong account of which approach is most externally valid — that is, which measurement approach most accurately captures how people behave in the real world. Indeed, there is an argument in favor of asking about both outcomes. Epstein et al. (ref. [36]., p.5) write that the "spillover effect [asking both outcomes] suggests that the social media context—and the mindset that it produces—actively interferes with accuracy discernment." This seems substantively important as (a) sharing is a defining feature of social media platforms, and (b) our manipulation is about truth discernment, so showing the positive effect of social corrections when social motivations are induced seems to strengthen our findings. Given our data, we ultimately cannot fully resolve questions implied by Epstein et al[36]. about whether our experimental estimates of user corrections would be different when only asking about a single outcome (and whether experimental outcomes might be different across the two outcomes). While it is not clear that an alternative design would greatly change the magnitude, direction, or significance of experimental estimates of user correction effects, we fully appreciate that others may disagree and that it remains an open empirical question. Future research can help clarify this.

Future research should also continue to investigate further boundary conditions and underlying mechanisms of social correction measures. Finally, on the applied side, how best to mobilize social media users to write corrective comments remains an open question[13].

## Conclusions
In three pre-registered cross-country experiments, we find that social corrections reduce perceived accuracy of and engagement with false news posts. We did not find a consistent pattern of statistically significant differences between our correction and miscorrection conditions with varying levels of signal amplification. Likewise, we did not find evidence that effects vary in a statistically significant way by anti-expert sentiments, cognitive reflection capacities, or susceptibility to social influence. We show that social miscorrections that flag *true* news as *false* likewise decrease perceived accuracy of and engagement with true news posts. While our results support the general effectiveness of social correction, they also suggest that miscorrections may cause further confusion on social media platforms.

## Data availability
All shareable data are found on the online OSF repository at https://osf.io/jhwfg.

## Code availability
All the code including reproducible analyses are found on the online OSF repository at https://osf.io/4hjcf for data relating to the UK, https://osf.io/yvdj4 for Italy, and https://osf.io/jhwfg Germany.

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

## Acknowledgements

This project received funding from the British Academy (BA Award SRG20\200348) and from Davidson College. The funders had no role in study design, data collection and analysis, decision to publish or preparation of the manuscript.

## Author contributions

FS and SS jointly led the conceptualization of the study design and manuscript preparation. FS led the data collection and project administration. SS led the data analysis and methodology. BL and JR contributed to the study design. BL, JR, SS, FS, and BC contributed to the writing. CR and BC contributed to the preparation of treatment materials.

## Competing interests

The authors declare no competing interests.
