## [Peer Review File · Communications Psychology]

21st Jun 23

Dear Dr Stoeckel,

Thank you for your patience during the peer-review process. Your manuscript titled "The double-edged sword of user comments: social corrections reduce the perceived accuracy of both false and real news in the UK, Germany, and Italy" has now been seen by 3 reviewers, and I include their comments at the end of this message. They find your work of interest, but raised some important points. We are interested in the possibility of publishing your study in *Communications Psychology*, but would like to consider your responses to these concerns and assess a revised manuscript before we make a final decision on publication.

We therefore invite you to revise and resubmit your manuscript, along with a point-by-point response to the reviewers. Please highlight all changes in the manuscript text file.

Editorially, we consider it important that you address Reviewer 2's request for additional analysis as well as expand your discussion of your findings in light of the work by Epstein et al., 2023. Please ensure that the limitations of your work are transparently discussed in a section titled "Limitations" in the Discussion.

Further processing of your manuscript will be greatly aided if the revision fully complies with our formatting and policy requirements. We therefore ask that you ensure that you use this checklist: <https://www.nature.com/documents/commspsychol-style-formatting-checklist-article-rr.pdf> to prepare your revision and ensure that the work complies with all requests. We advise you to upload your analysis code in a doi-minting repository already and include the link in the Code availability statement. We also very strongly recommend depositing the data that you will share publicly upon publication at this stage and that you include a reference to the deposition in the Data Availability statement. Sharing of the numerical data underlying the Figures is mandatory upon publication, while public sharing of the full anonymized data is strongly recommended but not required.

A common reason for delay is that statistics reporting and interpretation does not align with our guidelines. You will find more information on that in the checklist. We highlight in particular that where you report null findings, please make sure to use appropriate language to describe the results. (There is no statistical test that can demonstrate absence of an effect. Statements such as 'There is no difference between x and y.' or 'X does not affect Y.' must be revised to read 'We found [no/little] credible evidence of a difference between x and y.' or 'We found [no/little] credible evidence that X affects Y.')

Please use the following link to submit your revised manuscript, point-by-point response to the referees' comments (which should be in a separate document to any cover letter) and the completed checklist:

[link redacted]

Please do not hesitate to contact me if you have any questions or would like to discuss these revisions further. We look forward to seeing the revised manuscript and thank you for the opportunity to review your work.

Best regards,

Jennifer Bellingtier

Jennifer Bellingtier, PhD
Senior Editor
Communications Psychology

EDITORIAL POLICIES AND FORMATTING

Editorial Policy: [Policy requirements](https://www.nature.com/documents/nr-editorial-policy-checklist.pdf) (Download the link to your computer as a PDF.)

Furthermore, please align your manuscript with our format requirements, which are summarized on the following checklist:

[Communications Psychology formatting checklist](https://www.nature.com/documents/commspsychol-style-formatting-checklist-article-rr.pdf)

and also in our style and formatting guide [Communications Psychology formatting guide](https://www.nature.com/documents/commspsychol-style-formatting-guide-accept.pdf) .

*** TRANSPARENT PEER REVIEW:** Communications Psychology uses a transparent peer review system. This means that we publish the editorial decision letters including Reviewers' comments to the

authors and the author rebuttal letters online as a supplementary peer review file. However, on author request, confidential information and data can be removed from the published reviewer reports and rebuttal letters prior to publication. If your manuscript has been previously reviewed at another journal, those Reviewers' comments would not form part of the published peer review file.

* **CODE AVAILABILITY:** All Communications Psychology manuscripts must include a section titled "Code Availability" at the end of the methods section. In the event of publication, we require that the custom analysis code supporting your conclusions is made available in a publicly accessible repository; at publication, we ask you to choose a repository that provides a DOI for the code; the link to the repository and the DOI will need to be included in the Code Availability statement. Publication as Supplementary Information will not suffice. We ask you to prepare code at this stage, to avoid delays later on in the process.

* **DATA AVAILABILITY:**

All Communications Psychology manuscripts must include a section titled "Data Availability" at the end of the Methods section or main text (if no Methods). More information on this policy, is available at <http://www.nature.com/authors/policies/data/data-availability-statements-data-citations.pdf>.

At a minimum the Data availability statement must explain how the data can be obtained and whether there are any restrictions on data sharing. Communications Psychology strongly endorses open sharing of data. If you do make your data openly available, please include in the statement:

We recommend submitting the data to discipline-specific, community-recognized repositories, where possible and a list of recommended repositories is provided at <http://www.nature.com/sdata/policies/repositories>.

If a community resource is unavailable, data can be submitted to generalist repositories such as <https://figshare.com/> or <http://datadryad.org/> Dryad Digital Repository. Please provide a unique identifier for the data (for example a DOI or a permanent URL) in the data availability statement, if possible. If the repository does not provide identifiers, we encourage authors to supply the search terms that will return the data. For data that have been obtained from publicly available sources, please provide a URL and the specific data product name in the data availability statement. Data with a DOI should be further cited in the methods reference section.

REVIEWERS' EXPERTISE:

Reviewer #1 social media, misinformation

Reviewer #2 social media, misinformation

Reviewer #3 social media, misinformation

REVIEWERS' COMMENTS:

Reviewer #1 (Remarks to the Author):

This paper analyzes the role user comments on (mis)informative social media posts, examining whether and how such comments influence the perceived accuracy of the post, engagement with the post, and whether personality characteristics (anti-expert sentiment, cognitive reflection, and susceptibility to social influence) moderate these effects. Using a set of pre-registered experiments across three countries (UK, Italy, and Germany), the authors find a double-edged effect: social corrections do indeed lower perceived accuracy of misinformative posts, but social MIScorrections also lower the perceived accuracy of factually true posts. These effects are robust to the personality characteristics studied.

There is much to like about this article. Its use of pre-registration and the comparative aspect are much appreciated. I think with light revisions, this article would be suitable for publication in *Communication Psychology*. Below are my suggestions, which I hope the authors find useful when revising the manuscript.

First, some of the social corrections in the treatments are ambiguous. For example, the correction in the COP26 treatment says that there is something “deeply deeply fishy.” But in the conditions without a link, it’s not clear what is fishy: the conference’s behavior or the post criticizing the conference. The same is true for the Ukraine bombing condition, where the social correction simply says “What utter hogwash.” But it’s unclear whether that comment is meant to criticize Kiev’s alleged behavior or to call the veracity of the post into question. Fortunately, most of the treatments don’t suffer from this ambiguity, but nevertheless, the manuscript would benefit from more discussion about these comments.

Next, the description of the Germany study in Footnote 2 is confusing: “(2) the high amplification condition is similar to the lower amplification condition, but with a source cue (media outlet).” Why is the only difference between the high and low conditions a source cue? Shouldn’t it be the level of engagement? Furthermore, how does this source cue differ from the website provided in the link condition?

Finally, the text and labels in Figure 3 are so small that it is very difficult to discern what’s going on in those charts.

Reviewer #2 (Remarks to the Author):

In this paper, the authors investigate whether social correction affects the perceived accuracy of posts and engagement (liking intention and sharing intention). Moreover, the authors check if anti-

expert sentiments, cognitive reflection capacities, and susceptibility to social influence moderate the effect of social corrections.

Overall, the authors found that social correction decreases the perceived accuracy and engagement of false posts, and this effect also applies to true posts. There is no moderation effect of anti-expert sentiments, cognitive reflection capacities, and susceptibility to social influence.

The paper is interesting, well-organized, and sheds light on the role of social corrections.

I have one main concern and some questions and clarifications below.

The main concern I have is related to the design. I have some issues with asking for both accuracy and sharing (plus liking) intention. I think this raises a problem: one influences the other.

Epstein et al. (2023), found that participants were worse at discerning truth from falsehood if they both evaluated accuracy and indicated their sharing intention.

I'm not convinced about asking for all three "variables" in your design. Therefore, I believe this could affect your results and should be carefully addressed. And I think this is true even if you are testing experimentally some stimuli.

Below I list some other points.

I was wondering why participants were exposed to a different number of posts in two out of three studies, with a higher number of false posts (UK: 6 false vs. 3 true - Germany: 3 false vs. 6 true - Italy: 5 false vs. 4 true). Overall, false news constitutes a small proportion of news consumed online. Thus, I was wondering why you did not reverse the proportion or at least present the same number of false and true posts to participants.

Another question I have is why the order of the posts was fixed and not completely random. This is the case for the study in the UK; I'm not sure about Italy and Germany.

I was also wondering whether the social correction (the comments) was taken from real posts posted online. The same question for the number of reactions even if I don't think you took them online, given that you manipulated the number of reactions. Could you please confirm whether you informed participants that the comments (social corrections) were taken from real comments posted online? I think this could deceive participants.

Moving on to the analysis part, you could also run a linear regression where you regress, not only the conditions but also the veracity of the posts and the interactions between the conditions and the veracity of the posts.

I would also suggest stating more clearly in the introduction the design (around line 34).

I think the research idea is interesting, and I believe that social correction is an important aspect to study and explore.

Reviewer #3 (Remarks to the Author):

I enjoyed reading the paper. It is well-powered, has data from three countries, and the results follow

from the data. There are a few studies now suggesting, like this one, that interventions decrease not only beliefs in false ideas, but true ones as well. This paper, however, seems to do it better than the others I have read.

I am generally supportive of this paper being published. Some comments:

1. There is an epistemology issue underlying the results that should be touched on more: True and False ideas do not have inherent qualities that make them discernible on their own. People cannot only make these calls if they know something about the particular idea in advance, or something about the world in advance of exposure. So, there is no reason why interventions wouldn't also decrease the adoption of true ideas.
2. There is no reason to think that just because an idea is in a mainstream news source that it is true. There are false ideas everywhere, even in mainstream journalistic sources.
3. There is a normative issue here as well: What should people believe and when? Should people believe every NYT headline they see, or be more discerning by withholding belief for more evidence? Maybe getting people to believe fewer things, as the authors find of this treatment here, is a good thing, and not a bad thing. Regardless, this comes back to my point #1, that true and false ideas are not inherently different in their appearance simply because of their underlying truth value, because usually people cannot see the underlying truth of the matter directly.

Response to Referees — COMMSPSYCHOL-23-0129-T

Dear Referees,

Thank you very much for the opportunity to revise and resubmit this manuscript. We greatly appreciate your feedback, which we believe has allowed us to significantly improve the manuscript. Below, referee comments appear in gray and our responses appear in black.

Thank you very much for considering the revised version of our manuscript. We hope that, after reviewing our revisions, you will find the manuscript acceptable for publication in *Communications Psychology*. We are looking forward to hearing from you.

Reviewer #1 (Remarks to the Author):

This paper analyzes the role user comments on (mis)informative social media posts, examining whether and how such comments influence the perceived accuracy of the post, engagement with the post, and whether personality characteristics (anti-expert sentiment, cognitive reflection, and susceptibility to social influence) moderate these effects. Using a set of pre-registered experiments across three countries (UK, Italy, and Germany), the authors find a double-edged effect: social corrections do indeed lower perceived accuracy of misinformative posts, but social MIScorrections also lower the perceived accuracy of factually true posts. These effects are robust to the personality characteristics studied.

There is much to like about this article. Its use of pre-registration and the comparative aspect are much appreciated. I think with light revisions, this article would be suitable for publication in Communications Psychology. Below are my suggestions, which I hope the authors find useful when revising the manuscript.

First, some of the social corrections in the treatments are ambiguous. For example, the correction in the COP26 treatment says that there is something “deeply deeply fishy.” But in the conditions without a link, it’s not clear what is fishy: the conference’s behavior or the post criticizing the conference. The same is true for the Ukraine bombing condition, where the social correction simply says “What utter hogwash.” But it’s unclear whether that comment is meant to criticize Kiev’s alleged behavior or to call the veracity of the post into question. Fortunately, most of the treatments don’t suffer from this ambiguity, but nevertheless, the manuscript would benefit from more discussion about these comments.

1.1

We agree that some corrective comments are more directly and explicitly correcting the content of an original post than others. We made a specific design choice to use real world user comments in our stimuli. We opted for this approach not only for ethical reasons (less deception), but also to increase the external validity of our fieldwork. Fortunately, this choice afforded us the opportunity to (mostly) utilize corrective comments explicitly indicating that an original post is substantively incorrect. That our effects holds even though some of these comments are less explicit corrections than others, in our view, adds robustness to our findings. We added text to the section Materials to highlight this point:

“We increased external validity of the content used in the fieldwork by showing user comments that were actually written and posted by users on social media (albeit they might be from other posts than the ones shown). As a result, some corrections are more directly indicating that an original post is substantively inaccurate than others. For instance, the comment that there is “something deeply deeply fishy about this” might be perceived as a correction or in fact as agreement with the content of the original content. Most comments are direct and explicit corrections of the original post, e.g.: “This post is disingenuous if not a straight forward lie”. (This is a

corrective comment in False News Stimulus 8 from the UK; see osf for all materials.)”

Next, the description of the Germany study in Footnote 2 is confusing: “(2) the high amplification condition is similar to the lower amplification condition, but with a source cue (media outlet).” Why is the only difference between the high and low conditions a source cue? Shouldn’t it be the level of engagement? Furthermore, how does this source cue differ from the website provided in the link condition?

1.2

Thank you for this comment. We agree that the labeling of the conditions for the German study was imperfect (as we had an operationalization with source cues for true news). In practice, the German study does not have a low amplification condition and instead has two different versions of a high amplification condition -- one in which the underlying source featured in the social media post is clear, and one in which this source cue is absent. In the relevant figure, we now show two separate estimates for “high amplification” in Germany, one with source cue and one without source cue (rather than clumsily labeling the version without a source cue “low amplification”):

The revised figure looks like this:

False news

True news

Parameter estimates

country ● Germany ▲ Italy ■ UK ○ Germany (cue)

Note that we included the source cues manipulation to consider different factors that could affect how people process social media posts, and to enhance external validity where possible. For this reason, we used the Germany experiment to randomize source cues for the true news stimuli.

We thank the reviewer for pushing us to clarify these points, and to improve the presentation of results.

Finally, the text and labels in Figure 3 are so small that it is very difficult to discern what's going on in those charts.

1.3

We agree with your concern. Figure 3 is meant to show that we do not find heterogeneous treatment effects (i.e., the effect of a social correction on perceived accuracy is not moderated by anti-expert sentiments, cognitive reflection, and social influence). To improve readability, we not only enlarge the lettering, but also break out the plot into more legible components. Specifically, we now provide a separate figure for false and true news (see Supplementary Figure 1 and 2). We are happy to work with the editor to further refine our figures as necessary if accepted for publication.

Reviewer #2 (Remarks to the Author):

In this paper, the authors investigate whether social correction affects the perceived accuracy of posts and engagement (liking intention and sharing intention). Moreover, the authors check if anti-expert sentiments, cognitive reflection capacities, and susceptibility to social influence moderate the effect of social corrections.

Overall, the authors found that social correction decreases the perceived accuracy and engagement of false posts, and this effect also applies to true posts. There is no moderation effect of anti-expert sentiments, cognitive reflection capacities, and susceptibility to social influence.

The paper is interesting, well-organized, and sheds light on the role of social corrections.

2.1

Thank you!

I have one main concern and some questions and clarifications below.

The main concern I have is related to the design. I have some issues with asking for both accuracy and sharing (plus liking) intention. I think this raises a problem: one influences the other.

Epstein et al. (2023), found that participants were worse at discerning truth from falsehood if they both evaluated accuracy and indicated their sharing intention.

I'm not convinced about asking for all three "variables" in your design. Therefore, I believe this could affect your results and should be carefully addressed. And I think this is true even if you are testing experimentally some stimuli.

2.2

We thank the reviewer for pushing us on this point. Without one or more well powered auxiliary studies, it is true we cannot fully resolve questions implied by Epstein et al. (2023) about whether our experimental estimates of user corrections would be different when only asking about a single outcome (accuracy or sharing) rather than asking about both outcomes as we do here (and whether experimental outcomes might be different across the two outcomes). We reiterate our position that we do not think that this would greatly change the magnitude, direction, or significance of experimental estimates of user correction effects, though we fully appreciate that others may disagree and that it remains an open empirical question. Our hope is that these questions can remain for future research, rather than being a point for us to tackle in the current manuscript. We agree that we can highlight this more, so we have updated the discussion section to highlight this point as something that future research can help clarify.

We believe there are several reasons why advancing the manuscript as it currently stands is preferable to delaying until potential additional studies are done. 1) First and foremost, while

Epstein et al. (2023) demonstrate the measurement properties vary based on whether one or both outcomes are asked, there is not a strong account of which approach is most externally valid -- that is, which measurement approach most accurately captures how people behave in the real world. (To the extent there is an answer, we think it may favor asking about both outcomes. Epstein et al. write “This spillover effect [asking both outcomes] suggests that the social media context—and the mindset that it produces—actively interferes with accuracy discernment.” This seems substantively important as (a) sharing is an omnipresent feature of social media platforms, and (b) our manipulation is about truth discernment, so showing the positive effect of social corrections when social motivations are induced seems a feature rather than a bug.) 2) As research budgets are not infinite, we are faced with a tough choice that is hard to answer without evidence on external validity -- which outcome(s) do we ask about, or do we need to effectively run separate studies that are the same except with different outcome variables? (And if we choose only one, reviewers can make a similar critique that we did not choose the correct outcome to focus on or ask about both outcomes as both are substantively important and we only have one.) 3) While our conjecture is that the differences in the causal effect of our manipulations across different presentations of outcome variables is zero, we would need some credible prediction of what the expected moderating effect of outcome question format would plausibly be (from those that think there would be an effect) in order to conduct a power analysis that would allow us to make sure that we are sufficiently powered to reliably detect any hypothesized differences. We reiterate that we feel the best way to address these concerns in the current manuscript is in the discussion as a signpost for future research.

We added the following paragraph to the text:

“Additionally, our design asks respondents about both accuracy and sharing intent. Importantly, Epstein et al. (2023) found that participants were worse at discerning truth from falsehood if they both evaluated accuracy and indicated their sharing intention. However, while Epstein et al. demonstrate that measurement properties vary based on whether one or both outcomes are asked, there is not a strong account of which approach is most externally valid --- that is, which measurement approach most accurately captures how people behave in the real world. Indeed, there is an argument in favor of asking about both outcomes. Epstein et al. (p.5) write that the “spillover effect [asking both outcomes] suggests that the social media context—and the mindset that it produces—actively interferes with accuracy discernment.” This seems substantively important as (a) sharing is a defining feature of social media platforms, and (b) our manipulation is about truth discernment, so showing the positive effect of social corrections when social motivations are induced seems to strengthen our findings. Given our data, we ultimately cannot fully resolve questions implied by Epstein et al. (2023) about whether our experimental estimates of user corrections would be different when only asking about a single outcome (and whether experimental outcomes might be different across the two outcomes). While it is not clear that an alternative design would greatly change the magnitude, direction, or significance of experimental estimates of user correction effects, we fully appreciate that others may disagree and that it remains an open empirical question. Future research can help clarify this.

Below I list some other points.

I was wondering why participants were exposed to a different number of posts in two out of three studies, with a higher number of false posts (UK: 6 false vs. 3 true - Germany: 3 false vs. 6 true - Italy: 5 false vs. 4 true). Overall, false news constitutes a small proportion of news consumed online. Thus, I was wondering why you did not reverse the proportion or at least present the same number of false and true posts to participants.

2.3

At the beginning of the project, we debated whether to field identical studies across countries simultaneously, or whether we should run the studies sequentially in order to update our design as we proceeded. There are, of course, advantages and disadvantages to each approach. We elected for the latter. While the design in broad outline is consistent across all three studies, we did make adjustments along the way. One such adjustment was on the ratio of false and real news. We started with field work in the UK. We originally focused primarily on the role of social corrections in the context of false news. Real news items were mainly meant to present respondents with a more diverse set of social media posts (i.e., to avoid only showing false content). While we were always interested in the effect of corrections on real news, our interest in this grew as the initial UK fieldwork showed the double edged nature of corrective user comments. We decided to allocate more survey space to examine whether there were differential effects of corrective user comments across both false and real news in the subsequent fieldwork. Given survey length constraints, we could not show more than 9 posts and thus included an almost even number of false and real news posts in Italy (our judgment for choosing an odd number -- nine -- rather than a smaller even number -- eight -- was that statistical power was more valuable than an exactly even mix, a decision others could disagree with). In Germany, we implemented an additional design change in the context of real news. We tested if source cues of real news affect accuracy ratings (in the sense that miscorrections do not matter when source cues are displayed). Thus, the role of miscorrections in the context of real news was an even more prominent part of our field work in Germany, which we did last. This is why we decided to collect a larger number of real news accuracy ratings than false news ratings in Germany.

Setting aside these design decisions, we are also confident that the ratio of false and real news used in each study should not have influenced the outcomes. The authors of a recent study (Altay et al., 2023, <https://doi.org/10.31234/osf.io/t9r43>) conclude that “in the test of misinformation interventions (Altay, 2022; Guay et al., 2022), the ratio of falsity used in discernment tasks matters [...] only at extremes” (e.g., an 83% ratio of false to true may induce a more conservative response bias as detected using ROC analyses). In two studies, for instance, the authors found that “the ratio of false news participants were exposed to at the beginning of the experiment had no statistically significant effect on the accuracy ratings of subsequent true news [...] and subsequent false news.”

Another question I have is why the order of the posts was fixed and not completely random. This is the case for the study in the UK; I'm not sure about Italy and Germany.

2.4

Below, you can find information on the sequence in each of the countries. Indeed, in the UK, and to some extent also in Italy, the order is somewhat fixed. We randomize within false news, as respondents see a randomly chosen set of false news stories. In the German field work, respondents also see a randomly selected set of false news, in addition to randomly selected real news. Even in Germany, we did however use a set order rather than full randomization, i.e., real news posts come first, then respondents see a set of false news, and then they see again a set of real news.

We made several key design choices that made programming a fully randomized version of all the stimuli we created difficult. First, we made the design choice that all respondents would receive both true and false stimuli (and that they would see a true stimulus first). Second, any individual respondent was only shown a subset of the full set of false stimuli that we created. Third, we did not want respondents to receive stimuli in separate and uninterrupted blocks of either true or false content. We accept that there may be programming (javascript or otherwise) that could better randomize order with these constraints on the Qualtrics platform. As our primary strengths do not lie programming javascript add-ons, we made some simplifying choices to present stimuli in ways that address our goals and constraints. Given the smaller number of real news posts in the first study (UK), we fixed the real news and used randomization only for false news. We then kept this structure (fixed sequence of real and false news, but randomization within these blocks) for the two subsequent studies.

UK:

1 real news - 3 randomly selected false news - 1 real news - 3 randomly selected false - 1 real news

Italy:

1 real news - 2 randomly selected false news - 1 real news - 1 randomly selected false news - 1 real news - 2 randomly selected false news - 1 real news

Germany:

3 randomly selected real news - 3 randomly selected false news - 3 randomly selected real news

We thank the reviewer for giving us the opportunity to clarify text on this point. After the sentences that cover the general structure, we added a new sentence that makes it clear that there is randomization within a fixed sequence. For the British fieldwork, for example, we wrote:

“Six of the nine posts were false news posts, whereas the other three posts were true news posts. All participants were presented with the following sequence: (1) one true news post, (2) three false news posts (random subset of 10 posts), (3) one true news post, (4) three false news posts (random subset of 10 posts), and (5) one true news post. After seeing every post, respondents were asked to assess the (1) accuracy of, (2) probability to “like”, and (3) probability to share the post. Thus, there is randomization but also a fixed sequence (in all countries) which ensures that all respondents see false as well as true news.”

We have added similar explanations in the manuscript for the other two studies.

I was also wondering whether the social correction (the comments) was taken from real posts posted online. The same question for the number of reactions even if I don't think you took them online, given that you manipulated the number of reactions. Could you please confirm whether you informed participants that the comments (social corrections) were taken from real comments posted online? I think this could deceive participants.

2.5

The comments that we show respondents are actual comments that were posted by social media users. In some instances, we took corrective comments from other posts (i.e., posts not included in our fieldwork) and used them in the context of our posts. We make this choice, because as reviewer 1 points out, user corrections attached to false news online are not in all cases straight forward or clear in their goal. Thus, while we wanted to maximize external validity by using real comments, we wanted to make sure that most corrections are clear in their goal and specifically meant to correct the accuracy of a post. We did manipulate the number of corrective comments.

We told respondents “In the next section, we will show you a series of social media posts as they would appear on Twitter or Facebook.” We do not believe that this statement deceives respondents. We do want to note that we showed respondents a careful debriefing which lists all posts that were false and we offered links to a fact checking site with debunking information for each of them.

Moving on to the analysis part, you could also run a linear regression where you regress, not only the conditions but also the veracity of the posts and the interactions between the conditions and the veracity of the posts.

2.6

Thank you for this suggestion. Although we did not preregister these analyses, we now provide the results of the suggested analysis (we provide the link to our osf repository with regression outputs and marginal effects plots). We do not find an interaction between our treatment with veracity type (false vs. true news).

I would also suggest stating more clearly in the introduction the design (around line 34).

2.7

We added the following to the section following line 34 (we still kept this somewhat short in order to avoid repeating details that are covered around in the Methods section).

“Each participant rated a set of nine social media posts after answering a set of pre-treatment questions. Respondents either saw social media posts without user comments (control condition) or one of three treatment conditions that included user comments, some of which denote the original post as inaccurate.”

I think the research idea is interesting, and I believe that social correction is an important aspect to study and explore.

2.8

Thank you very much for your expertise and support!

Reviewer #3 (Remarks to the Author):

I enjoyed reading the paper. It is well-powered, has data from three countries, and the results follow from the data. There are a few studies now suggesting, like this one, that interventions decrease not only beliefs in false ideas, but true ones as well. This paper, however, seems to do it better than the others I have read.

I am generally supportive of this paper being published. Some comments:

1. There is an epistemology issue underlying the results that should be touched on more: True and False ideas do not have inherent qualities that make them discernible on their own. People cannot only make these calls if they know something about the particular idea in advance, or something about the world in advance of exposure. So, there is no reason why interventions wouldn't also decrease the adoption of true ideas.

3.1

Based on this helpful feedback, we have revised our justification for studying social miscorrections as follows:

“Third, Bode and Vraga (2022) warn that social miscorrections of true news might amplify the spread of misinformation, at least in some cases. Specifically, they find that when social media users flag factually accurate information—e.g., tick bites can trigger an allergy to red meat—as incorrect, people are less likely to believe this information. This finding is in line with the argument that there are no inherent differences between true and false claims themselves, and their accuracy depends on their coherence with the real world rather than built-in linguistic markers – which requires audiences to bring pre-existing knowledge to bear (but see e.g, Carrasco-Farre, 2022). Along these lines, we might expect social miscorrections to just as readily distort public understanding of facts. Clearly, we need to further probe this potential for corrections to have negative effects when they are implemented on true news regardless of intent.”

2. There is no reason to think that just because an idea is in a mainstream news source that it is true. There are false ideas everywhere, even in mainstream journalistic sources.

3.2

We agree with the reviewer that a source cue does not necessarily indicate that a piece of news is true (and articles in mainstream news might indeed be inaccurate). What we test is a heuristic that social media users might use: namely that a source cue (from a mainstream media outlet) affects how users balance the perceived veracity of a post with user comments. Specifically, people might see a piece of information with a mainstream source cue as more likely to be true than a similar piece of information without a mainstream source cue.

Added to true news text (main body):

“The reason for showing source cues in the original posts in Germany is to test if respondents use it as a heuristic for accuracy that makes miscorrections irrelevant (even if main stream news can be inaccurate as well). Yet, the treatment effects show that miscorrections affect users in spite of source cues.”

3. There is a normative issue here as well: What should people believe and when? Should people believe every NYT headline they see, or be more discerning by withholding belief for more evidence? Maybe getting people to believe fewer things, as the authors find of this treatment here, is a good thing, and not a bad thing. Regardless, this comes back to my point #1, that true and false ideas are not inherently different in their appearance simply because of their underlying truth value, because usually people cannot see the underlying truth of the matter directly.

3.3

We thank the reviewer for pushing us harder on considering the normative implications of our findings. As researchers, we are personally uncertain what the optimum “Goldilocks” level of skepticism is at the micro, meso, or macro levels. We do share the underlying sentiment that levels of information skepticism can be too low as well as too high. We have tried to incorporate this comment by providing a more nuanced discussion of skepticism in our revised Discussion:

“The simplicity of creating effective corrections is however a double-edged sword. Social media environments also include users that flag true news as false, and these instances can exacerbate confusion in the public sphere. Although skepticism towards reputable news can also be important (Li, 2023), recent work has highlighted the fact that not all skepticism is healthy, and can cross a line into cynicism or anti-expert sentiment (Li, 2023; Chinn & Hasell, 2023). Accordingly, given the trade-off between desirable and undesirable effects of social correction measures, it is important to reflect on the relative volume of false content shared and the accuracy of “corrective” commentary. The vast majority of news shared on social media comes from high credibility sources, rather than dubious domains (e.g., Guess et al., 2019). Other descriptive work suggests that for at least one news story (President Trump’s approval rating in 2017), up to 20% of accompanying social commentary was false (Anspach & Carlson, 2020)...”

11th Sep 23

Dear Dr. Stoeckel,

Your manuscript titled "The double-edged sword of user comments: social corrections reduce the perceived accuracy of both false and real news in the UK, Germany, and Italy" has now been seen by our reviewers, whose comments appear below. In light of their advice I am delighted to say that we are happy, in principle, to publish a suitably revised version in Communications Psychology under the open access CC BY license (Creative Commons Attribution v4.0 International License).

We therefore invite you to revise your paper one last time to address the remaining concerns of our reviewers and a list of editorial requests. At the same time we ask that you edit your manuscript to comply with our format requirements and to maximise the accessibility and therefore the impact of your work.

EDITORIAL REQUESTS:

SUBMISSION INFORMATION:

OPEN ACCESS:

Communications Psychology is a fully open access journal. Articles are made freely accessible on publication under a [CC BY license](http://creativecommons.org/licenses/by/4.0) (Creative Commons Attribution 4.0 International License). This license allows maximum dissemination and re-use of open access materials and is preferred by many research funding bodies.

For further information about article processing charges, open access funding, and advice and support from Nature Research, please visit <https://www.nature.com/commspsychol/article-processing-charges>

At acceptance, you will be provided with instructions for completing this CC BY license on behalf of all authors. This grants us the necessary permissions to publish your paper. Additionally, you will be asked to declare that all required third party permissions have been obtained, and to provide billing

information in order to pay the article-processing charge (APC).

* **DATA AVAILABILITY:**

[link redacted]

Best regards,

Jennifer Bellingtier

Jennifer Bellingtier, PhD
Senior Editor
Communications Psychology

REVIEWERS' EXPERTISE:

Reviewer #1 social media, misinformation

Reviewer #2 social media, misinformation

Reviewer #3 social media, misinformation

REVIEWERS' COMMENTS:

Reviewer #1 (Remarks to the Author):

I appreciate the time the authors took to engage with the suggestions of the reviewers. I think this is a great contribution to the field and I think it should be accepted for publication.

Reviewer #2 (Remarks to the Author):

I thank the authors for addressing all my comments.

In the spirit of fostering a constructive debate and considering my interest in the topic, I would like to share a thought I've had with the authors (please notice, this is a sharing of thoughts). In your study, there is a negative effect of social correction when it is implemented on true news. Indeed you found that social corrections that flag true news as false decrease the perceived accuracy of and engagement with true news posts. From my perspective, reducing the spread of misinformation requires using different interventions together (see Bak-Coleman et al. 2022), because it seems difficult for one single intervention could solve the problem. Therefore, it would be interesting to explore the interplay between social correction and other interventions.

Reviewer #3 (Remarks to the Author):

The authors have addressed my comments appropriately.

Response to Referees — COMMSPSYCHOL-23-0129A

Dear Referees,

We greatly appreciate your feedback, which we believe has allowed us to significantly improve the manuscript in the last round of revisions. We briefly reply to the specific comments of the reviewers in the latest round.

We have revised the manuscript as well as the supplementary information substantially in order to incorporate the editorial requests.

Reviewer #1 (Remarks to the Author):

I appreciate the time the authors took to engage with the suggestions of the reviewers. I think this is a great contribution to the field and I think it should be accepted for publication.

Response:

We are delighted that reviewer 1 is pleased with our revisions of the manuscript. We are grateful for the kind words on our study.

Reviewer #2 (Remarks to the Author):

I thank the authors for addressing all my comments.

In the spirit of fostering a constructive debate and considering my interest in the topic, I would like to share a thought I've had with the authors (please notice, this is a sharing of thoughts). In your study, there is a negative effect of social correction when it is implemented on true news. Indeed you found that social corrections that flag true news as false decrease the perceived accuracy of and engagement with true news posts. From my perspective, reducing the spread of misinformation requires using different interventions together (see Bak-Coleman et al. 2022), because it seems difficult for one single intervention could solve the problem. Therefore, it would be interesting to explore the interplay between social correction and other interventions.

Response:

We are pleased to hear that we were able to address all of this reviewer's comments. We also appreciate the thought that the reviewer mentioned. We did not revise the paper further since the reviewer flagged this specifically as a thought rather than an issue we need to address in the paper. Indeed, it is very likely that it requires a combination of strategies and tools to combat misinformation on social media. Examining the interplay of different interventions seems like a most fruitful as well as important field for further study.

Reviewer #3 (Remarks to the Author):

The authors have addressed my comments appropriately.

Response:

We are happy to hear that we have been able to address this reviewer's comments.